# Insights into pulmonary phosphate homeostasis and osteoclastogenesis emerge from the study of pulmonary alveolar microlithiasis

Yasuaki Uehara [1] ✉, Yusuke Tanaka[1], Shuyang Zhao [2], Nikolaos M. Nikolaidis[1], Lori B. Pitstick[1], Huixing Wu[1], Jane J. Yu[1], Erik Zhang[1], Yoshihiro Hasegawa[1], John G. Noel[1], Jason C. Gardner[1], Elizabeth J. Kopras [1], Wendy D. Haffey[3], Kenneth D. Greis [3], Jinbang Guo[4], Jason C. Woods [4], Kathryn A. Wikenheiser-Brokamp[5], Jennifer E. Kyle [6], Charles Ansong[6], Steven L. Teitelbaum [7], Yoshikazu Inoue[8], Göksel Altinişik [9], Yan Xu [2,10] ✉ & Francis X. McCormack [1] ✉

Pulmonary alveolar microlithiasis is an autosomal recessive lung disease caused by a deficiency in the pulmonary epithelial Npt2b sodium-phosphate co-transporter that results in accumulation of phosphate and formation of hydroxyapatite microliths in the alveolar space. The single cell transcriptomic analysis of a pulmonary alveolar microlithiasis lung explant showing a robust osteoclast gene signature in alveolar monocytes and the finding that calcium phosphate microliths contain a rich protein and lipid matrix that includes bone resorbing osteoclast enzymes and other proteins suggested a role for osteoclast-like cells in the host response to microliths. While investigating the mechanisms of microlith clearance, we found that Npt2b modulates pulmonary phosphate homeostasis through effects on alternative phosphate transporter activity and alveolar osteoprotegerin, and that microliths induce osteoclast formation and activation in a receptor activator of nuclear factor-κB ligand and dietary phosphate dependent manner. This work reveals that Npt2b and pulmonary osteoclast-like cells play key roles in pulmonary homeostasis and suggest potential new therapeutic targets for the treatment of lung disease.

Pulmonary alveolar microlithiasis (PAM) is an autosomal recessive disorder (OMIM 265100) caused by mutations of the SLC34A2 gene, which results in deficiency of the Npt2b sodium-phosphate co-transporter[1]. As surfactant phospholipids are catabolized by alveolar macrophages, phosphate accumulates in the alveolar lining fluid due to loss of the epithelial pump. Calcium phosphate crystals form spontaneously in the lumen of pulmonary alveolar airspaces,

ultimately resulting in pulmonary fibrosis and pulmonary hypertension. Over 1000 PAM cases have been reported, mostly from Japan, Turkey, and Italy. The disease is slowly progressive, with most patients living into middle age, but lethality is reported in all age groups including infants[2,3]. There are currently no effective therapies for PAM. Therapeutic whole lung lavage and etidronate therapy have been attempted with mixed and generally disappointing results[4-9], and

current management options are limited to supportive care or lung transplantation.

The role of microliths in the pathogenesis of PAM have not been well studied. Barnard reported that microliths are composed of calcium and phosphate in ratios that are consistent with hydroxyapatite[10]. In our previous study, we reported that dietary phosphate restriction decreased alveolar microlith accumulation in a PAM animal model, but the mechanisms involved remained unclear[11]. In this study, we found upregulation of multiple osteoclast genes and proteins in PAM lung, as well as in the bronchoalveolar lavage cells and the pulmonary parenchyma of Npt2b[−/−] mice, including those for signature osteoclast proteins tartrate-resistant acid phosphatase (TRAP), ATPase H+ transporting V0 subunit d2 (ATP6V0D2), calcitonin receptor (CALCR) and cathepsin K (CTSK). Intratracheal adoptive transfer of isolated microliths into Npt2b[+/+] mice induced a nearly identical genetic signature as well as bone resorbing capacity in alveolar macrophages (AM) isolated by bronchoalveolar lavage (BAL) one week later. Both recruited monocytes and tissue resident alveolar macrophages contributed to microlith clearance. Dietary phosphate restriction augmented osteoclast-related gene expression in the lung, and reduced the burden of microliths through receptor activator of nuclear factor-κB ligand (RANKL)-dependent osteoclast-like multinucleated giant cell (MNGC) differentiation. High dietary phosphate levels increased microlith burden in the Npt2b[−/−] lung by elevating levels of phosphate and osteoprotegerin, the soluble RANKL decoy receptor, in the alveolar space. In contrast, Npt2b[+/+] mice were able to maintain alveolar phosphate balance through a range of dietary phosphate intake levels. These studies reveal previously unrecognized roles for Npt2b in pulmonary phosphate homeostasis and for myeloid lineage derived osteoclast-like cells in response to particulate challenge in the lung, both of which have broader translational implications.

## Results

### Scanning electron microscopy and elemental, proteomic and lipidomic analysis of human and mouse PAM microliths

To better understand the pathogenesis of PAM, we first analyzed the composition of microliths isolated from the lungs of Npt2b[−/−] mice and a PAM patient. Scanning electron microscopy (SEM) imaging revealed spherical particulate microliths with a hairy, fibrous coating, particularly for the human stones, and human and mouse microlith diameters that ranged between 100–500 µm (Fig. 1a, b) and 5–20 µm (Fig. 1c, d), respectively. The sodium dodecyl sulfate (SDS) washed mouse microliths had a porous, spongy, cortical bone-like surface (Fig. 1e–g) that was similar to that of the synthetic hydroxyapatite microspheres (Fig. 1h, i). Elemental analysis of the mouse and human microliths by energy-dispersive spectroscopy identified calcium and phosphorus in ratios that were consistent with hydroxyapatite, but slightly lower than that of synthetic hydroxyapatite crystals, due to incorporation of proteins and/or lipids into the matrix of the native stones (Supplementary Table 1). Proteomic analyses of microliths from both species identified over 300 proteins including those commonly found in renal stones, such as alpha-2HS-glycoprotein, prelamin-A/C and annexin A2, and in the alveolar lumen, such as surfactant proteins A, B, C and D (Fig. 1j, Supplementary Data 1 and 2). Osteoclast proteins including osteopontin (OPN), CTSK and TRAP were also detected in the mouse microliths. Lipidomic analyses of the stones revealed the presence of surfactant phospholipids, in proportions that were roughly commensurate with their abundance in pulmonary surfactant (Fig. 1k, Supplementary Table 2, Supplementary Data 3).

### Osteopontin expression is increased in PAM lung

There is considerable interest in OPN as a potentially useful pulmonary biomarker, especially in idiopathic pulmonary fibrosis and other idiopathic interstitial pneumonias which exhibit marked expansion of SPP1 positive alveolar macrophages compared to controls[12–15]. To evaluate

OPN expression in PAM lung and its potential as a biomarker, OPN levels in BAL fluid (BALF) and serum were determined. OPN levels in BALF were significantly higher in Npt2b[−/−] mice than Npt2b[+/+] mice (Fig. 1l). Furthermore, serum OPN levels in Npt2b[−/−] mice and the PAM patients were significantly elevated compared to Npt2b[+/+] littermates or human controls, respectively (Fig. 1m, n). Immunohistochemical staining demonstrated OPN expression localized in monocyte/macrophages present around the microliths in the PAM patient lung (Fig. 1o), and to some extent in alveolar and airway epithelial cells. Single-cell analysis of the lung of a PAM child demonstrated that >95% of the cells expressing SPP1 in the PAM lung were AM (see below). In contrast, only faint OPN staining was present in AM and airway epithelium of control lungs.

### Osteoclast-related gene signatures are upregulated in PAM lung

The finding that bone degrading enzymes produced by osteoclasts (such as TRAP and CTSK) were incorporated into the matrix of microliths led us to consider a potential role for osteoclast-like cells in microlith clearance in the PAM lung. We employed single-cell RNA-seq (scRNAseq) analysis on a lung explant from a PAM child with a homozygous SLC34A2 mutation in c. 524-18_559del and a control lung, and displayed the results in a uniform manifold approximation and projection (UMAP) plot in Fig. 2a. Twenty-three distinct cell populations were identified by an unbiased approach using the Human Lung Atlas and the SingleR machine learning algorithm (see Methods), representing epithelial (alveolar type I (AT1), alveolar type II (AT2), Club/basal, ciliated, neuroendocrine), endothelial (capillary types 1 and 2, venous, bronchial, lymphatic), mesenchymal (fibroblasts and smooth muscle), and immune cells (alveolar macrophages and proliferating macrophages, classical and nonclassical monocytes and dendritic cells, T cells, B cells, mast cells, plasma cells). Expression profiles of known markers were used to validate cell type assignments (Supplementary Fig. 1). Transcriptomic profiles of macrophages from the PAM child and a normal donor were extracted and integrated by a mutual nearest neighbor batch correction algorithm, and displayed on a UMAP plot, splitting PAM and normal donor for clarity (Fig. 2b). Expression of selected osteoclast markers in the monocyte/macrophage clusters is shown on feature plots in Fig. 2c. Integrated and split UMAPs showing expression of classical alveolar macrophage markers reveals that 'normal AM' populations persist despite robust osteoclast-like differentiation (Supplementary Fig. 2). A heatmap illustrates upregulation of expression of key osteoclast-related genes in PAM lung macrophages compared to healthy control macrophages (Fig. 3a) including *ATP6VOD2*, osteopontin (*SPP1*), and dendrocyte expressed seven transmembrane protein (*DCSTAMP*), among other genes (Supplementary Data 4). A dot plot depicting both the expression frequency and level of expression of osteoclast genes is consistent with robust osteoclast differentiation of AM, with concomitant reduction in expression of the AM marker gene, *EMR1* (Fig. 3b). Other genes included in the plot were RANKL mediated osteoclastogenesis signaling genes; GRB2 associated binding protein 2 (*GAB2*), nuclear factor of activated T cells 1 (*NFATC1*), melanocyte inducing transcription factor (*MITF*), TNF receptor associated factor 6 (*TRAF6*) and inhibitor of nuclear factor kappa B kinase subunit beta (*IKBKB*), (b) osteoclast cell fusion related genes; *DCSTAMP* and osteoclast stimulatory transmembrane protein (*OCSTAMP*), (c) osteoclast precursor specific genes; tartrate-resistant acid phosphatase (*ACP5*), transcription factor EC (*TFEC*), and integrin subunit beta 3 (*ITGB3*), and d) osteoclast receptors, enzymes and acid secretion genes; calcitonin receptor (*CALCR*), *CTSK*, and matrix metallopeptidase 9 (*MMP9*). Additional subclustering of macrophages identified 5 PAM subtypes and 2 control macrophage subtypes (Fig.3c). The expression of predicted macrophage markers across macrophage subtypes is shown in dot plots (Supplementary Fig. 3). The transcriptomic signature of PAM_Mac1 and CON_Mac1 overlapped and both expressed typical AM genes,

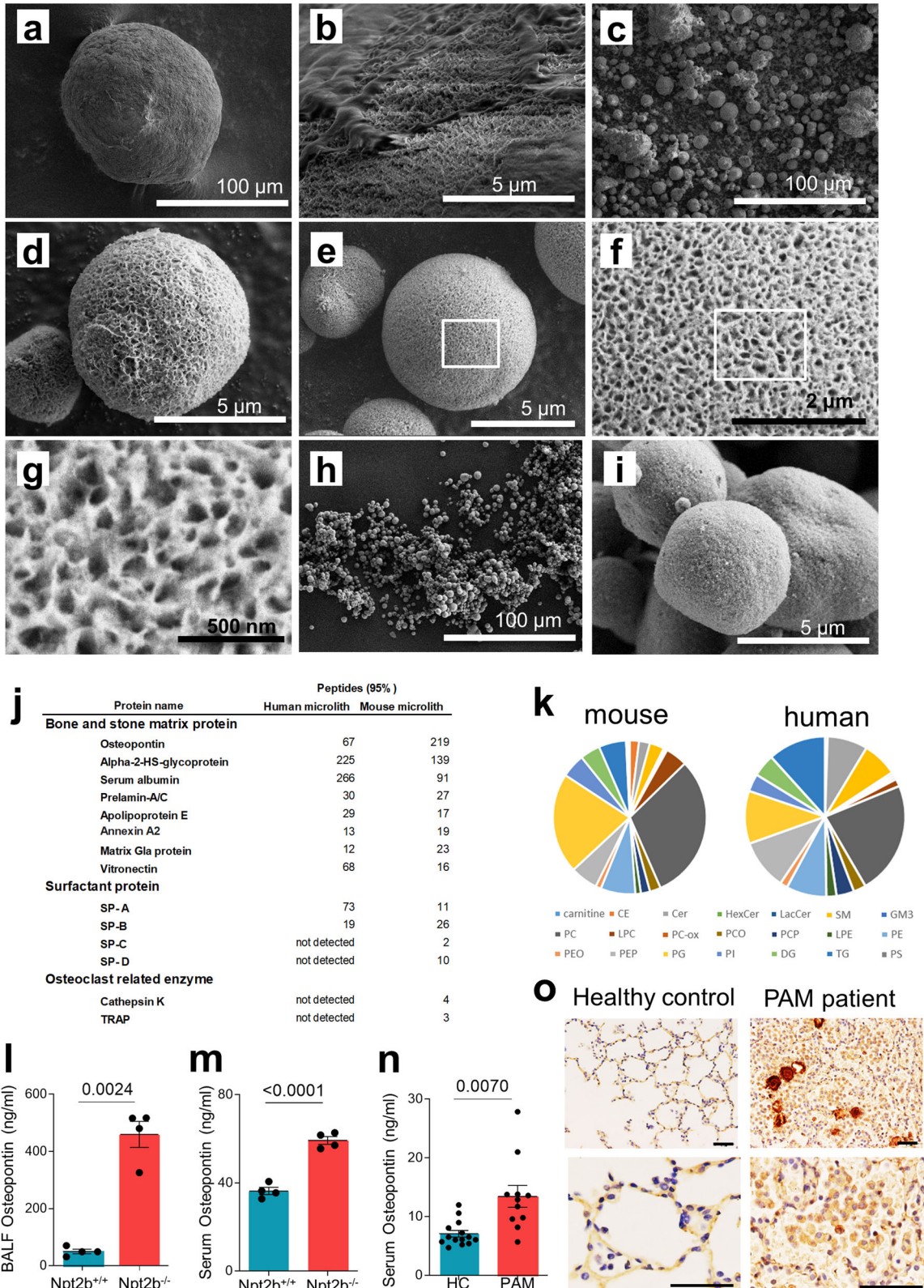

consistent with tissue resident AM. PAM_Mac2 and PAM_Mac3 were the most osteoclast-like, based on expression of multiple marker genes (*ACP5, DCSTAMP, RANK, MMP9, ATP6V0D2*). PAM_Mac4 and Con_Mac2 expressed classical monocyte markers such as S100A8 and CD14 and were negative for osteoclast markers and CD115 low, suggesting origins as recruited monocytes. Expression of top selected osteoclast markers is shown on a dot plot in Fig. 3d. Pseudotime analysis suggests

a trajectory of differentiation from monocytes to PAM_Mac3 through PAM_Mac4, which has a gene expression pattern that is intermediate between monocytes and PAM_Mac3 (monocyte-macrophage fusion) (Fig. 3e). PAM_Mac4 also serves as a potential progenitor for other macrophage branches, PAM_Mac1,2,5. Comparisons of the expression of various osteoclast markers between macrophage subtypes and multiple other immune cell types is shown in Supplementary Fig 4.

**Fig. 1 | Microliths are composed of hydroxyapatite, have a porous bony surface, and contain bone matrix proteins, and alveolar components, surfactant proteins and phospholipids. a–i** Scanning electron microscopy (SEM) of microliths from a human infant lung explant (**a**, **b**), from Npt2b⁻/⁻ mice (**c**, **d**) and commercially available hydroxyapatite spheres (**h**, **i**). Microliths isolated from Npt2b⁻/⁻ mice washed with 0.02% SDS at low, medium and high power (**e**–**g**). Proteins (**j**) and lipids (**k**) detected in microliths from Npt2b⁻/⁻ mice and PAM patient respectively. Osteopontin quantified by ELISA in mouse BAL fluid (**l**) and serum (*n* = 4) (**m**) and serum from PAM patients (PAM) (*n* = 14) and healthy controls (HC) (*n* = 11) (**n**). IHC for osteopontin in human control and PAM human infant (**o**). Scale bars, 50 μm.

Data are expressed as means ± SD. *P* values shown in charts were determined by unpaired two-tailed t-test with Welch correction (**l**–**n**). Source data are provided as a Source Data file. Abbreviations: SP-A(-B, -C, -D) surfactant protein A (B, C, D, respectively), TRAP tartrate-resistant acid phosphatase, PC phosphatidylcholine, PE phosphatidylethanolamine, PG phosphatidylglycerol, PI phosphatidylinositol, PS phosphatidylserine, SM sphingomyelin, PEO 1-o-alkyl-2-acyl-PE, CE cholesterol ester, LPC lysophosphatidylcholine, PEP PE plasmalogen, Cer ceramide, PC-ox oxidized PC, Hex-Cer hexosylceramide, PCO 1-o-alkyl-2-acyl-PC, Lac-Cer lactosylceramide, PCP PC plasmalogen, DG diacylglycerol, LPE lysophosphatidylethnolamine, TG triglyceride, GM3 monosialodihexosylganglioside.

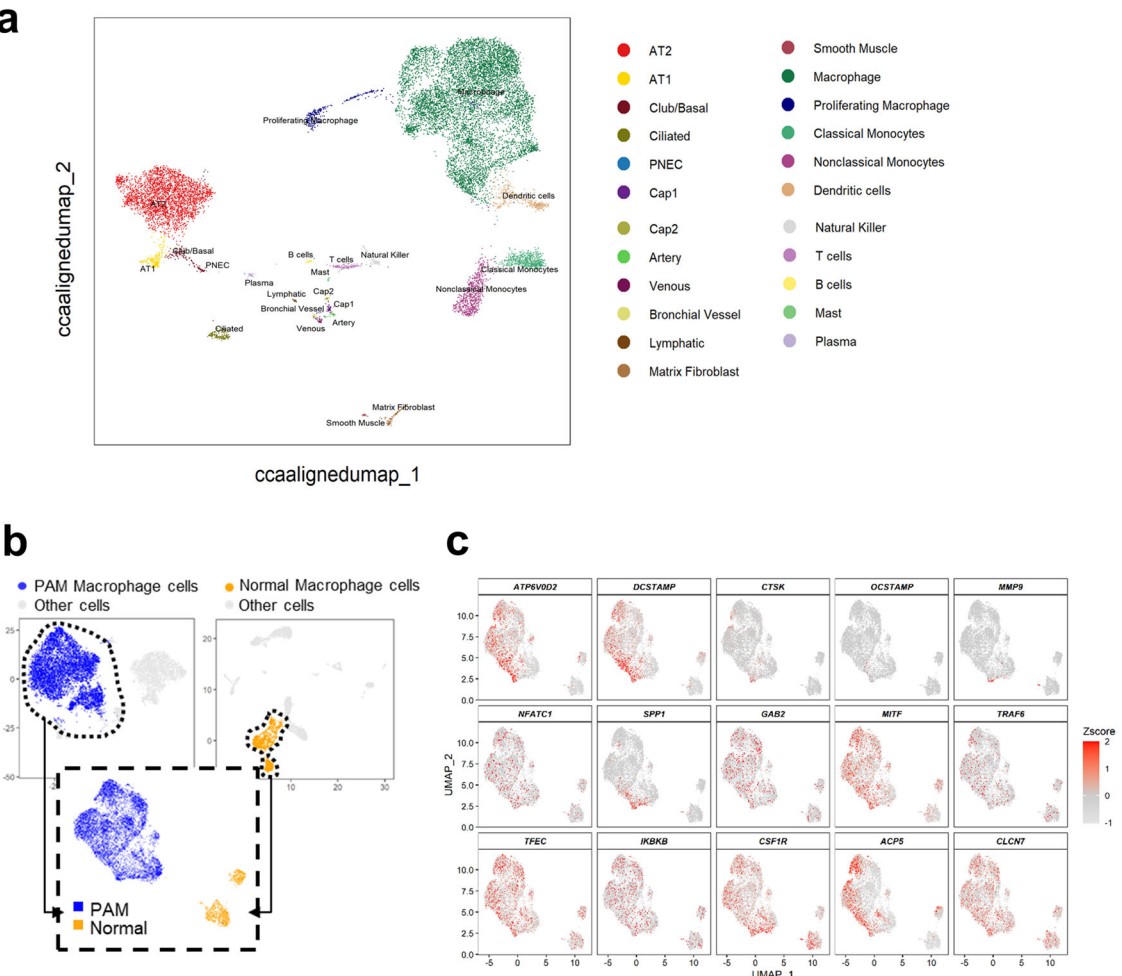

**Fig. 2 | Integrated Single-cell RNA-seq analysis identifies diverse cell populations. a** Uniform Manifold Approximation and Projection (UMAP) representation of 14,210 cells from PAM and control donor lung samples. Twenty-three cellular populations were identified, each dot represents a single cell and cells are color coded to represent cell identity. A reference-based machine learning cell type annotation algorithm (SingleR) was used to predict the cell populations. **b** Two thousand five hundred and thirty-four macrophage cells from the PAM sample and 1831 macrophage cells from the normal control sample were extracted and re-clustered. Cells are visualized by UMAP and colored by condition. **c** Expression of selected osteoclast-related genes is shown in UMAP. The expression is z-score normalized.

## Co-localization of TRAP and CTSK expressing multinucleated giant cells and microliths

We explored osteoclast gene expression in BAL cells from Npt2b⁻/⁻ mice, and found upregulation of *Acp5, Mmp9, Ctsk, Itgb3,* and *Calcr* (Fig. 4a–e). In addition, the gene expression of colony stimulating factor-1 (*Csf1*), a cytokine important for osteoclast differentiation, was upregulated in Npt2b⁻/⁻ BAL cells (Fig. 4f). TRAP, CTSK and CALCR-positive MNGCs were found in Npt2b⁻/⁻ mouse lungs (Fig. 4h, l, p) and the PAM patient lung (Fig. 4j, n, r), but only faint TRAP, CTSK CALCR staining was found in AMs of Npt2b⁺/⁺ (Fig. 4g, k, o) and healthy human

lung (Fig. 4i, m, q). Interestingly, the airway epithelium also stained positively for CALCR (Fig. 4o, q). Although most CALCR-positive myeloid cells in the PAM lung were mononuclear (Fig. 4r, s), many clustered in aggregates, presumably as a precursor to fusion into MNGCs (Fig. 4s). Many MNGCs colocalized with microliths and were often found adherent to microliths (Fig. 4j, n, s). The number of nuclei per MNGC and the size of MNGCs in human PAM lung tended to be greater than mouse MNGCs. The presence of TRAP and CTSK or CALCR-positive MNGC and mononuclear cells adherent to microliths was confirmed in two additional patients (Supplementary Fig. 5).

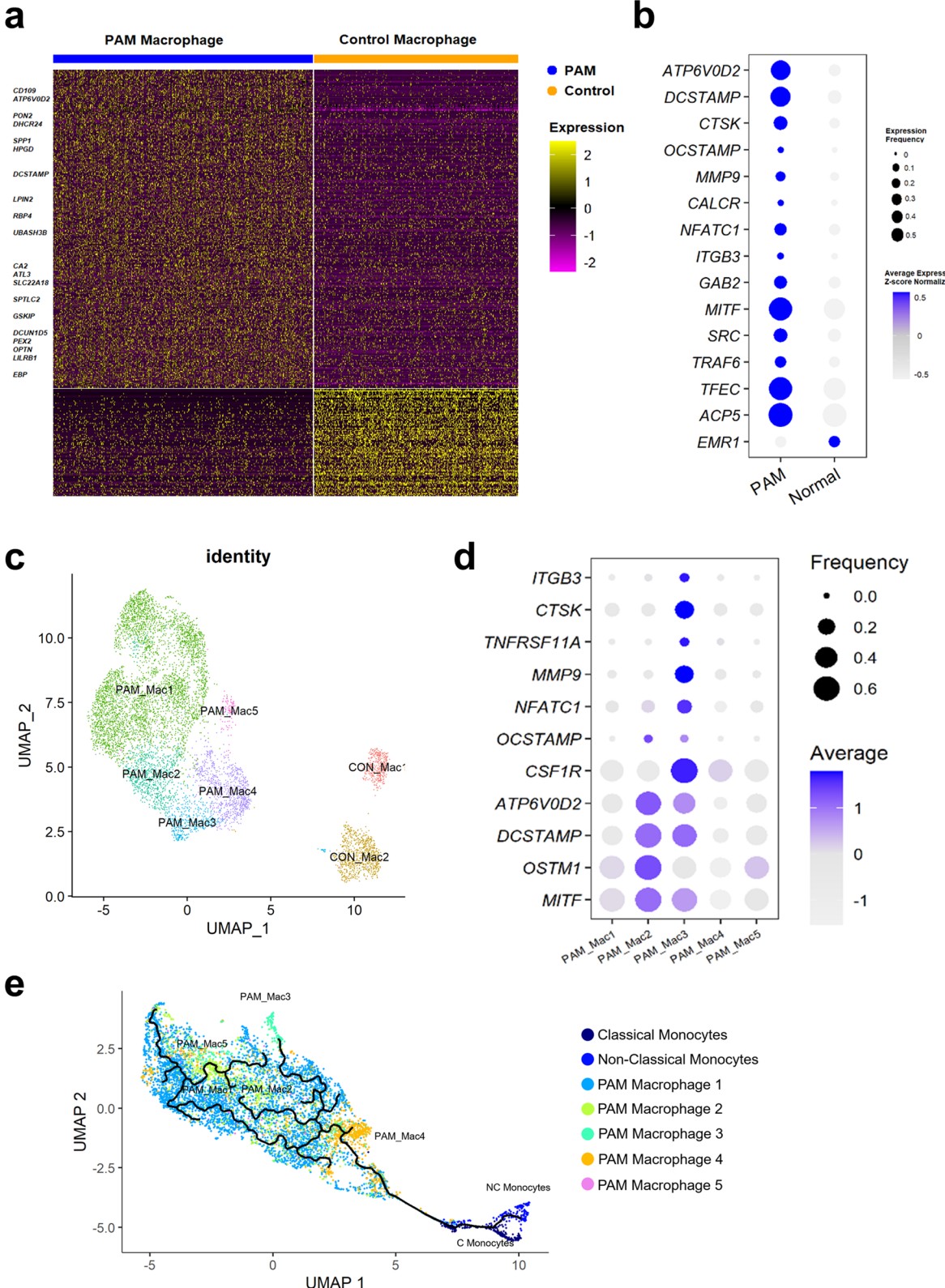

## Osteoclastic activities were upregulated in the lungs of Npt2b$^{-/-}$ mice lung and Npt2b$^{+/+}$ recipients of adoptively transferred hydroxyapatite microliths

CTSK activity in the lung was assessed using the in vivo imaging system (IVIS) with Cat K 680 (Cat K), a CTSK-cleavage activated probe. Fluorescence intensity of Cat K was increased in the lungs of Npt2b$^{-/-}$ mice lung compared to Npt2b$^{+/+}$ mice at 18 h post intratracheal Cat K injection (Fig. 4t, u). These results demonstrate marked upregulation

of an enzymatically active signature osteoclast protease (i.e CTSK) in the lungs of Npt2b$^{-/-}$ mice.

## Microlith clearance is delayed in CCR2$^{-/-}$ mice

Adoptive transfer of microliths into wild-type (WT) lungs induced *Acp5, Mmp9, Ctsk and Itgb3* genes expression in BAL cells at day 7 post challenge (Fig. 5a–d). TRAP positive monocytes and macrophages engulfed microliths and formed inflammatory aggregates

**Fig. 3 | Single-cell RNA sequencing reveals upregulation of osteoclast genes in lung macrophages from a PAM patient. a** Heatmap demonstrating the differentially expressed genes of lung macrophages from a PAM patient (PAM macrophage) vs lung macrophages from a healthy control (control macrophage). Differentially expressed genes were calculated using a binomial based test. Genes with FDR < 0.1, expression frequency > 20%, and the effective size (frequency ratio) > 2 were considered as differentially expressed. Selected osteoclast-related genes among the differentially expressed genes are labeled. **b** Dot plot showing selected osteoclast-related genes and their expression comparison between PAM macrophages and control macrophages. Node size represents expression frequency of gene expression in each condition. Node color represents the z-score scaled average gene expression in each group. Osteoclast-related genes were selected with modified criteria: *p*-value <0.05 and effect size or fold change >1.3. **c** scRNAseq data of 7781 macrophage cells from PAM and control lungs were used for this analysis. Macrophage cells were clustered using Jaccard-Louvain clustering algorithm, cells were visualized by UMAP and colored by macrophage sub-populations. **d** Dot plot showing the expression of osteoclast markers in the five predicted macrophage subtypes in PAM lung. **e** Monocle 3 was used to reconstruct a trajectory model for PAM monocytes and macrophages. The predicted trajectory started from monocytes →PAM-Mac4 (monocyte-macrophage fusion) →PAM-Mac3 (osteoclast-like). PAM-Mac4 also serves as a progenitor of other macrophage branches (PAM-Mac1, 2 and 5).

around microliths at day 7 post adoptive transfer, and both microliths (by von Kossa staining) and inflammatory changes (by H&E staining) were markedly diminished to absent by day 28 (Fig. 5e, left panels). To determine the role of tissue resident AM vs. monocyte derived macrophages in microlith clearance, we repeated the experiment in CCR2$^{-/-}$ mice which cannot recruit monocytes because the cells fail to exit the marrow[16]. After adoptive transfer of microliths into CCR2$^{-/-}$ mice, although in general the inflammatory trajectory closely paralleled that of the CCR2$^{+/+}$ mice, TRAP expression in inflammatory sites was diminished compared to CCR2$^{+/+}$ mice, and microlith clearance was delayed despite resolution of inflammation by day 28 (Fig. 5e, right panels). These results indicate that microliths induce osteoclastic transformation of BAL cells, and monocyte recruitment is required for optimal clearance of stones.

## Osteoclastogenesis and osteoclastic activation in BAL cells pretreated with hydroxyapatite cultured with RANKL

A bone resorption assay was used to determine if microliths induce this signature osteoclast function in BAL cells. As proof of concept, bone marrow derived monocytes incubated in the presence of M-CSF + RANKL but not M-CSF alone induced TRAP positive MNGC formation when cultured on plastic, and actin ring formation and bone pitting on bovine bone slices (Supplementary Fig. 6a). Similarly, d14 BAL cells isolated from mice challenged with synthetic hydroxyapatite microspheres, but not BAL cells from d14 saline challenged mice, exhibited TRAP positive MNGC formation (Fig. 5f–k) when cultured on plastic in the presence of M-CSF and low dose (25 ng/ml) RANKL (Fig. 5g, j). At higher doses of RANKL (100 ng/ml), even BAL cells from d14 saline treated mice and CCR2$^{-/-}$ mice produce some TRAP positive MNGC when cultured on plastic (Fig. 5h, and Supplementary Fig. 6b). In addition, actin ring formation (Fig. 5m), bone pitting (Fig. 5o) and release of bovine type I collagen C-telopeptide fragment, CTX-1, into the media (Fig. 5p) is seen to a greater extent in BAL cells isolated from d14 HA than d14 saline challenged mice (Fig. 5l, n) when cultured on bovine bone slices in the presence of M-CSF and RANKL (100 ng/ml). Collectively, these results indicate that i.t. hydroxyapatite challenge promotes the differentiation of BAL monocytes and AM into mature lung osteoclasts capable of degrading the mineral and osteoid components of bone.

## Low-phosphate diet reverses microlith accumulation in PAM model mice

To determine the effect of dietary phosphate intake on microlith accumulation, we fed Npt2b$^{+/+}$ and Npt2b$^{-/-}$ mice with low-phosphate diets (LPD; 0.1% phosphate), regular diets (RD; 0.7% phosphate) and high-phosphate diets (HPD; 2% phosphate) for a total of 8 weeks. The pre- and post-dietary treatment radiographs and micro CTs revealed that LPD reduced and HPD increased the profusion of calcific densities in all lung fields (Fig. 6a–c). Quantitative CT measures also demonstrated that LPD significantly decreased, and HPD tended to increase microlith burden relative to relevant age-matched controls (Fig. 6d, e for LPD and Fig. 6f and Supplementary fig 7 for HPD). Next we treated Npt2b$^{+/+}$ and Npt2b$^{-/-}$ mice with LPD, RD or HPD for 1 week to analyze

the effect of dietary phosphate intake on phosphate and calcium homeostasis. Serum parathyroid hormone (PTH) levels were higher on a regular diet at baseline in Npt2b$^{-/-}$ mice than in Npt2b$^{+/+}$ mice (Fig. 6g), and tended to be higher in PAM patients than healthy controls (Fig. 6h), but there were no other significant baseline differences from cognate controls in NPT2B deficient mice or humans in serum fibroblast growth factor 23 (FGF-23) (Figs. 6i, j), 1,25-dihydroxyvitamin D (VitD3) (Fig. 6k, l), phosphate (Fig. 6m, n) or calcium (Fig. 6o, p). LPD decreased serum phosphate, PTH and FGF-23 and increased serum VitD3 and serum and BAL calcium (Fig. 6g, i, k, m, o) in mice of both genotypes, but BAL phosphate and calcium levels were modulated by dietary phosphate intake (Fig. 6q, r) only in Npt2b$^{-/-}$ mice. *Slc20a1* (*Pit1*) and *Slc20a2* (*Pit2*) phosphate transporter gene expression was increased in isolated AT2 from LPD treated mice compared to RD or HPD treated animals in mice of both genotypes. (Fig. 6s, t). Xenotropic and polytropic retrovirus receptor 1 (*Xpr1*) phosphate transporter, transient receptor potential cation channel subfamily V member 6 (*Trpv6*) and S100 calcium binding protein G (*S100g*) calcium transporter gene expression were also increased by LPD compared to HPD in Npt2b$^{-/-}$ mice but not Npt2b$^{+/+}$ mice (Supplementary Fig. 8). These results indicate that phosphate and calcium levels in BAL from Npt2b$^{-/-}$ mice are sensitive to dietary phosphate content, and that the decrease of phosphate and calcium levels in alveolar space upon treatment with LPD may in part be the result of induction of alternative phosphate and calcium transporters in the lung.

## RANKL-dependent osteoclast activation is required for optimal microlith clearance

We found that 2 weeks of LPD treatment of Npt2b$^{-/-}$ mice decreased microlith burden without a quantitative change in TRAP positive MNGC accumulation in the lungs compared to RD treatment (Fig. 7a, b). Although HPD increased microlith burden over that seen with RD treatment, fewer TRAP positive cells were found in the lung. Consistent with this finding, osteoclast-related gene expression profiles in whole lung homogenates revealed lower levels for *Tnfrsf11a* (receptor activator of nuclear factor-κB (RANK)), *Acp5* and *Mmp9* as dietary phosphate intake increased (Fig. 7c–e). Osteoprotegerin (OPG), which acts as a sink for RANKL, was significantly elevated in BALF after HPD treatment compared to LPD or RD treatment (Fig. 7f) of Npt2b$^{-/-}$ mice, providing a plausible explanation for the decrease in RANKL-dependent osteoclast gene expression and TRAP positive MNGC formation that were found in the lungs of the HPD mice (Fig. 7a, b). The likely source of RANKL in the lung was determined to be AT2 cells, based on an 8-fold greater abundance of RANKL in AT2 cell lysates from Npt2b$^{-/-}$ mice compared to Npt2b$^{+/+}$ mice (Fig. 7g), and the lack of significant between group differences in RANKL in BAL cell lysates from the animals (Supplementary Fig. 9). RANKL expression was not well detected in the scRNAseq analysis, although a few RANKL expressing T lymphocytes and NK cells were present (Supplementary Fig. 4). Similarly, RANKL was at or below the limit of detection in the BALF, likely due to ~100-fold dilution of alveolar lining fluid inherent in this method (which also confounds detection of GM-CSF, another key alveolar macrophage cell differentiation factor). To confirm the

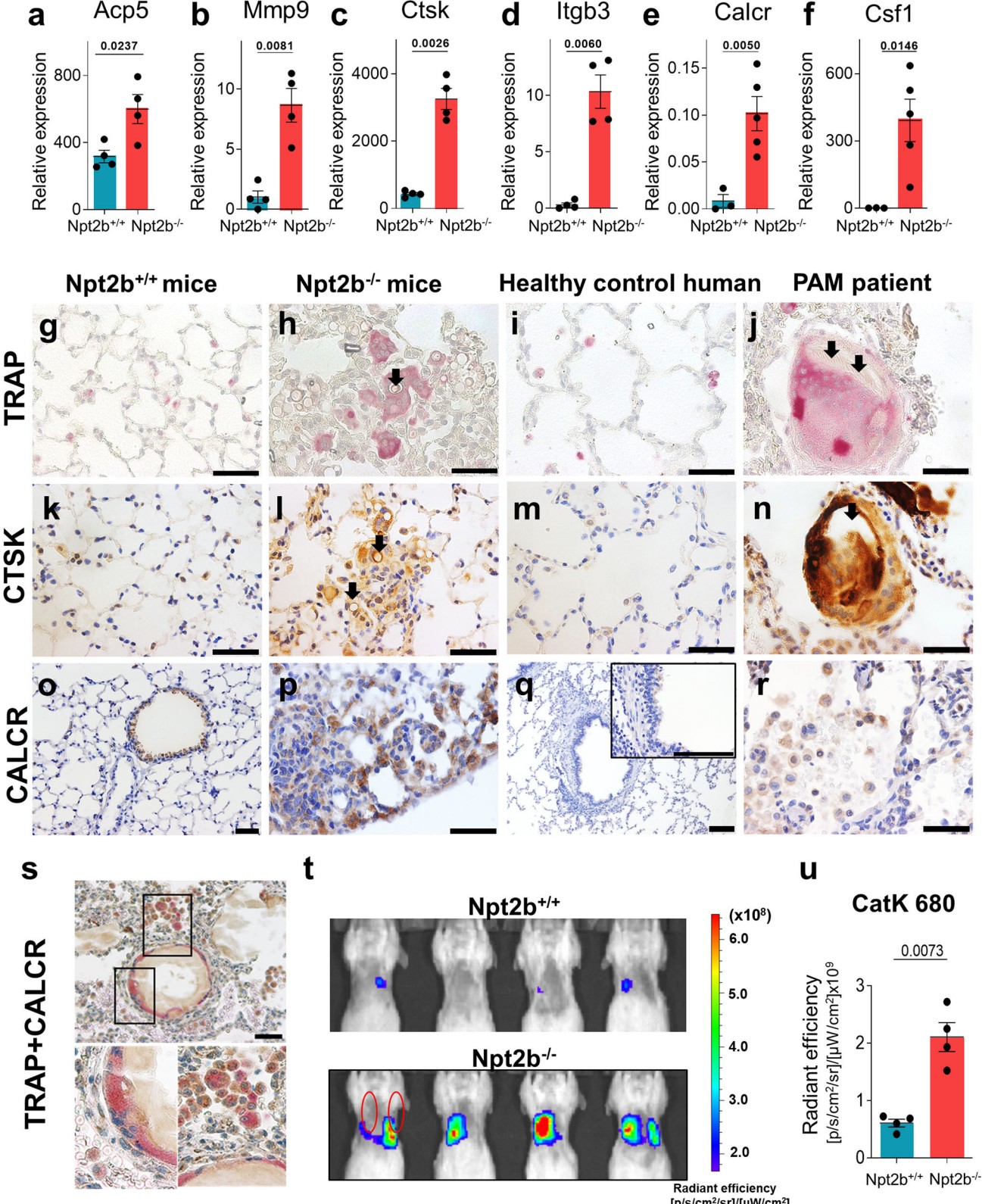

**Fig. 4 | Genetic, immunohistochemical, enzymatic and functional evidence of osteoclast activity in PAM lung. a–f** Relative expression of osteoclast-related genes in the BAL cells from 8- to 10-week-old Npt2b+/+ and Npt2b−/− mice were determined by RT-PCR. n = 4 mice per group (**a–d**), or n = 3 for Npt2b+/+ mice and n = 5 for Npt2b−/− mice (**e, f**). **g–s** Lung sections of 8- to 10-week-old Npt2b+/+ (**g, k, o**) and Npt2b−/− mice (**h, l, p**) and a human control (**i, m, q**) and PAM patient (**j, n, r, s**) were fixed and stained for TRAP (**g–j**), CTSK (**k–n**), CALCR (**o–r**) or co-staining of TRAP and CALCR (**s**). Scale bars, 50 μm. **t–u** Npt2b+/+ and Npt2b−/− mice were injected with the cleavage activated CTSK substrate, Cat K 680 FAST, and fluorescent images were acquired 18 h later by IVIS (**t**) and the fluorescent signal was quantified (**u**). n = 4 mice per group. Data are expressed as means ± SD. P values shown in charts determined by unpaired two-tailed t-test with Welch correction (**a–f, u**). Source data are provided as a Source Data file.

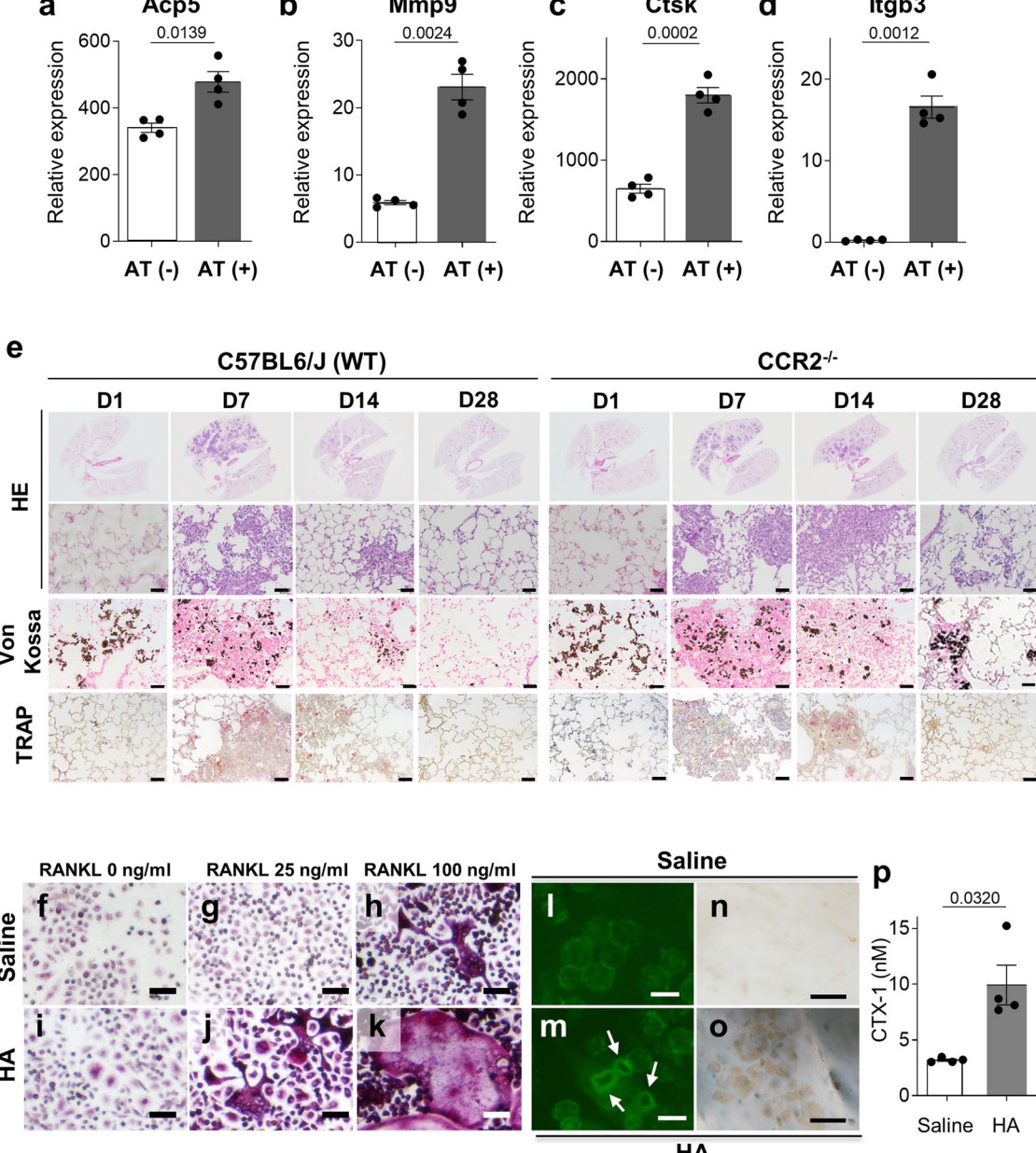

**Fig. 5 | Adoptive transfer of hydroxyapatite microliths induces osteoclast gene expression, functional activity and osteoclastogenesis in Npt2b+/+ mice.** Microliths isolated from the lungs of Npt2b−/− mice were adoptively transferred into the lungs of C57BL/6J (WT) mice (**a–d**) and C57BL/6J CCR2+/+/CCR2−/− mice (**e**). **a–d** BAL was performed on C57BL/6J mice at day 7 post microlith instillation, and expression of osteoclast-related genes in BAL cells was determined by RT-PCR. *n* = 4 mice per group. Data are expressed as means ± SD. **e** CCR2+/+ and CCR2−/− mice were sacrificed at the indicated time points after intratracheal microlith instillation, and the lungs were formalin fixed and stained with H&E, von Kossa, and TRAP reagent as indicated. Representative images are shown. Scale bar, 50 μm. **f–p** BAL cells isolated from C57BL/6J mice at day 14 post intratracheal challenge with saline (saline)

or hydroxyapatite (HA) were plated on plastic plates or bovine bone slices with M-CSF and RANKL at indicated concentrations for **f–k** or 100 ng/ml for **l–o**. At day 6, the cells cultured on the plastic plates were stained for TRAP activity (**f–k**). For BAL cells plated on bone slices, actin rings were visualized by phalloidin staining (white arrow) (**l, m**) and resorbed bone area was visualized by peroxidase-conjugated wheat germ agglutinin/horse-radish peroxidase staining after removing cells (**n, o**). Scale bars: 50 μm (**f–k, n, o**) and 20 μm (**l, m**). The levels of CTX-1 in bone culture medium were measured by ELISA (*n* = 4 wells per group) (**p**). *P* values shown in charts determined by unpaired two-tailed t-test with Welch correction (**a–d, p**). Data are expressed as means ± SD. Source data are provided as a Source Data file.

essential role of RANKL in osteoclast mediated clearance of microliths in PAM lung, we inhibited RANKL-dependent osteoclastogenesis in Npt2b−/− mice using a neutralizing anti-RANKL antibody. The fluorescent labeled bisphosphonate, Osteosense, was used to label microliths within the alveoli of living animals and assess the effect of the RANKL neutralization on microlith clearance. Osteosense was injected intratracheally into Npt2b−/− mouse lungs and the decay in the fluorescence signal emanating from the microliths was tracked serially

using IVIS for 12 days. Anti-RANKL neutralizing antibody treatment (3 times per week, i.p.) reduced the abundance of TRAP positive MNGCs (Fig. 7h, i) in histological sections and inhibited in vivo microlith clearance compared to control antibody treatment (Fig 7j, k). In addition, multinucleation and association of MNGC with microliths was altered; weakly TRAP positive mononuclear cells that were neither aggregated or attached to stones predominated in the lung sections of anti-RANKL treated Npt2b−/− mice (Fig. 7h). These results indicate that

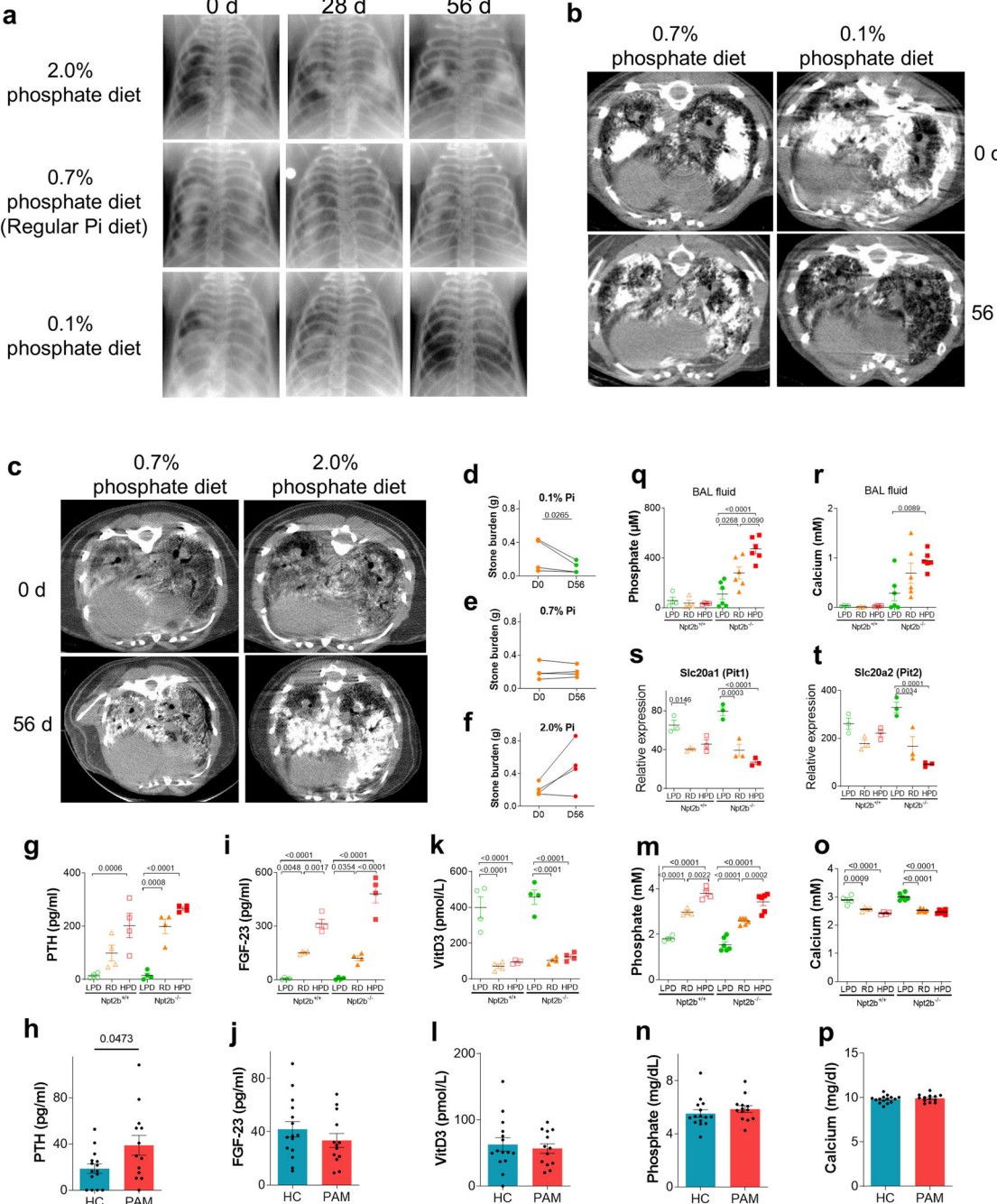

**Fig. 6 | Dietary phosphate modulates phosphate homeostasis and stone accumulation in Npt2b⁻/⁻ mice. a–f** Twenty- to 28-week-old Npt2b⁻/⁻ mice (**a**, **b**) or 5 to 6-week-old Npt2b⁻/⁻ mice (**c**) were fed with low (0.1%) (LPD), regular (0.7%) (RD) or high (2%) phosphate diet (HPD) for 8 weeks as indicated. **a** Serial radiographs of individual Npt2b⁻/⁻ mice after 0, 28 and 56 days on the indicated phosphate diet. **b** Serial microCT images of individual Npt2b⁻/⁻ mice before and after 56d on RD (0.7% phosphate (Pi)) and LPD (0.1% Pi) diet. **c** MicroCT images of Npt2b⁻/⁻ mice before and after 56 days on RD (0.7% Pi) and HPD (2.0% Pi) diet. **d–f** Stone burden in the lung at day 0 and day 56 calculated from CT images of 20–28-week-old mice treated with RD (0.7% Pi) and LPD (0.1% Pi) and 5–6-week-old mice treated with HPD. **g, i, k, m, o, q–t** Five- to six-week-old Npt2b⁺/⁺ and Npt2b⁻/⁻ mice were fed with 0.7% Pi diet for 2 weeks as pretreatment, then regular, low or high-phosphate diets (0.7% Pi (RD), 0.02% Pi (LPD) or 2% Pi (HPD), respectively) for 1 week. Serum and

BAL fluid were collected after treatment; PTH (**g**), FGF-23 (**i**), VitD3 (**k**), phosphate (**m**) and calcium (**o**) levels in serum were determined. Phosphate (**q**) and calcium (**r**) levels in BAL fluid were determined. Relative expression of phosphate transporters, *Slc20a1* (**s**) and *Slc20a2* (**t**) in the isolated AT2 cells from Npt2b⁺/⁺ and Npt2b⁻/⁻ mice fed with each indicated diet were determined by RT-PCR. N = 4 mice per group (**g**, **i**, **k**), n = 4 for Npt2b⁺/⁺ mice and n = 6 for Npt2b⁻/⁻ mice (**m**, **o**, **q**, **r**), or n = 3 mice per group (**s**, **t**). Serum was collected from PAM patients (PAM) (n = 15) and healthy controls (HC) (n = 13), and PTH (**h**), FGF-23 (**j**), VitD3 (**l**), phosphate (**n**) and calcium (**p**) levels in serum were quantified by ELISA. Data are expressed as means ± SD. *P* values shown in charts determined by two-tailed Ratio paired t test (**d–f**), unpaired two-tailed t-test with Welch correction (**h**, **j**, **l**, **n**, **p**), one-way ANOVA Tukey's multiple comparisons test (**g**, **i**, **k**, **m**, **o**, **q–t**). Source data are provided as a Source Data file.

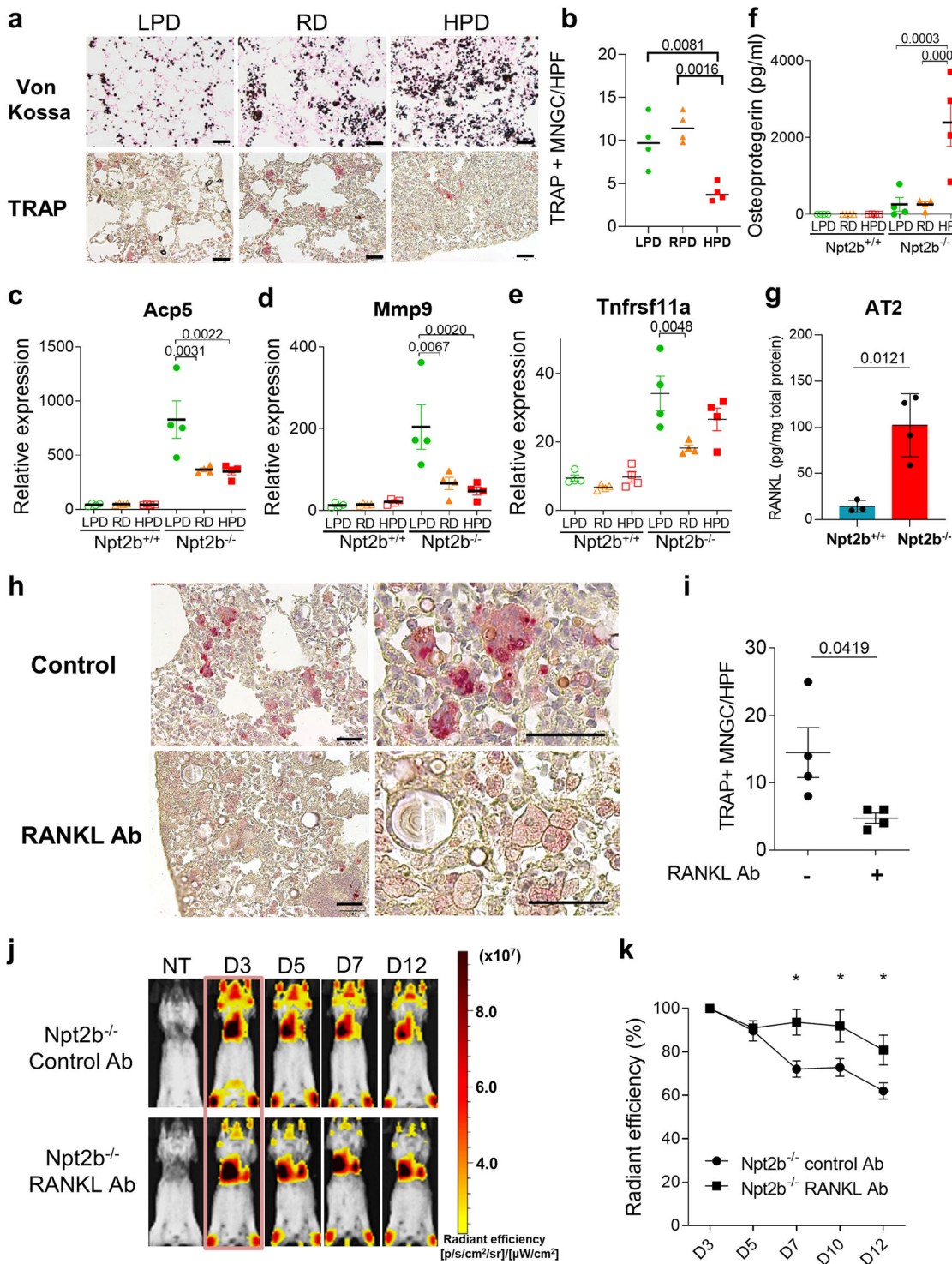

**Fig. 7 | RANKL-dependent osteoclast activity is required for microlith clearance. a, b** Five to six-week-old Npt2b[+/+] and Npt2b[−/−] mice were fed with regular 0.7% Pi diet (RD) for 2 weeks as pretreatment, then fed RD, 0.02% Pi (LPD) or 2% Pi (HPD). Von Kossa and TRAP staining of the lung after 2 weeks of RD, LPD and HPD treatment (**a**) and TRAP positive MNGCs were counted in 10 random high-power fields (HPF) (×100) and the average number was used for analysis. Scale bars, 50 µm (**b**). Relative expression of *Acp5* (**c**), *Mmp9* (**d**) and *Tnfsfr11a* (**e**) in the whole lung and osteoprotegerin levels (**f**) in BALF were determined by RT-PCR and ELISA respectively after 1 week of RD, LPD and HPD treatment. *n* = 4 mice per group. **g** RANKL levels in AT2 cell lysate from Npt2b[+/+] and Npt2b[−/−] mice were measured by ELISA. *n* = 3 for Npt2b[+/+] mice and *n* = 4 for Npt2b[−/−] mice per group. **h, i** Npt2b[−/−] mice were treated with the RANKL neutralizing antibody or a rat IgG2a isotype control IgG intraperitoneally for 28d, then TRAP staining was performed on the lung sections (**h**) and TRAP positive MNGCs were counted in 10 random HPF (×100) and the average number was used for analysis. Scale bars, 50 µm (**i**). *n* = 4 per group. **j, k** Npt2b[+/+] and Npt2b[−/−] mice were treated with the RANKL neutralizing antibody or a rat IgG2a isotype control IgG intraperitoneally for 7d, and then injected intratracheally with the OsteoSense 750EX probe (osteosense) to label microliths. Fluorescent reflectance imaging were acquired on the IVIS at the indicated times (before probe injection, 3, 5, 7 and 12 d after probe injection). *n* = 6 per group. Data are expressed as means ± SD. Compared to Npt2b[−/−] control Ab, left to right, **p* = 0.012, **p* = 0.0474 and **p* = 0.0367. P values shown in charts determined by one-way ANOVA Tukey's multiple comparisons test (**b−f**), unpaired two-tailed t-test (**g, i, k**). Source data are provided as a Source Data file.

RANKL-dependent osteoclast formation and actions are required for microlith clearance.

## Discussion

This study reveals osteoclast-like differentiation and activity in mononuclear cells of the PAM lung, and a role for these cells in limiting the accumulation of microliths in PAM. Rapid progression of hyperdense pulmonary infiltrates in a PAM patient placed on alendronate, a bisphosphonate known to inhibit osteoclast function, suggested a potential role for osteoclasts in modulating the stone burden in PAM[4,17]. The observation that the microliths from humans and mice contained osteoclast bone degrading proteins CTSK, CALCR and TRAP, was also consistent with this notion. Gene expression analyses by single-cell RNA-seq in a human PAM lung and RT-PCR in Npt2b[−/−] mouse lung revealed robust osteoclast signatures in pulmonary myeloid cells. Histologically, abundant TRAP and CTSK positive MNGCs were found attached to stones in mouse and human lung, and BAL cells isolated from mice that had been challenged with hydroxyapatite degraded bone when plated on slices of bovine femur. A low-phosphate diet prevented and reversed microlith accumulation by mechanisms that included down modulation of alveolar OPG expression, and upregulation of osteoclast activity and calcium and phosphate transporter expression. Inhibition of osteoclastogenesis with anti-RANKL antibody attenuated osteoclast gene expression, diminished TRAP and CTSK positive MNGC formation, and slowed the clearance of stones from the lungs of Npt2b[−/−] mice. Collectively, these data support a role for pulmonary osteoclast-like cells in pulmonary homeostasis for a rare disease, and raise the question of whether these newly described pulmonary cells may play a broader pathogenic or physiologic role in the lung.

Phosphate liberated by the catabolism of surfactant by AM is secreted into the epithelial lining fluid (ELF). The loss of functional Npt2b, due to inactivating mutations of the sodium-phosphate cotransporter gene, Slc34a2, results in a defect in epithelial phosphate transport and an increase in ELF phosphate (Fig. 6q). Calcium also increases in the ELF of the PAM lung by an unknown mechanism, perhaps simply by passive action to maintain electroneutrality, and complexes with phosphate leading to the accumulation of microliths in the lumen of the alveolus (Fig. 6r). We find no evidence of expression of the essential osteoblast transcription factor, osterix, in single-cell analysis or osteocalcin expression (arguably the most specific osteoblast marker in tissues) by IHC to suggest participation of bone forming cells in microlith formation. Analysis of microliths from PAM patients and Npt2b[−/−] mice reveals a bone-like, porous surface, a simple elemental composition limited to calcium and phosphate at a 5:3 ratio consistent with hydroxyapatite, and a matrix containing abundant proteins and lipid species[11]. The presence of surfactant proteins and lipids in the matrix of the microliths confirmed suspicions that they incorporate alveolar macromolecular components as had been suggested by their lamellar appearance on histological sections[11].

Osteoclasts are specialized cells derived from circulating hematopoietic cells of the monocyte/macrophage lineage[18]. Osteoclast differentiation typically occurs on the bone surface and progresses in stepwise manner from mononuclear myeloid progenitor to polykaryon to activated osteoclast[19]. Under the influence of RANKL and M-CSF, osteoclastogenic genetic programs are executed by transcription factors, PU.1, MITF, TRAF6, NFkβ and NFATc1 that drive the expression of proteins required for cell fusion, attachment to bone, and secretion of acid and proteases to degrade mineralized osseous tissues[20]. Membrane bound RANKL expressed on the plasma membrane of osteocytes is thought to be the major source of this primary osteoclastogenic cytokine, requiring cell to cell contact for engagement of the RANK receptor on the surface of osteoclast precursors, but soluble RANKL which is liberated by the action of proteases also plays an important role[21,22]. OPG produced by osteoblasts, epithelial cells, endothelial cells, dendritic cells and B cells serves as a sink for RANKL, by binding and neutralizing the cytokine to modulate osteoclastogenesis[23]. DCSTAMP, OCSTAMP and DNAX activation protein of 12 kDa (DAP12) are some of the effectors known to be required for the fusion of mononuclear progenitors, to yield multinuclear giant cells with up to 15–20 nuclei[24–26]. Cytoskeletal rearrangements form actin rings that create a zone of occlusion when ITGB3 on the osteoclast plasma membrane[27] binds to the asp-gly-xxx motif of OPN and other sialoproteins on bone. Activated osteoclasts secrete hydrochloric acid into the potential space that is formed between bone and cell, called Howship's lacunae, through active transport of $H^+$ and $Cl^-$ by acid pumps including ATP6V0D2 and chloride voltage-gated channel 7 (CLCN7), respectively, to degrade the mineral components of bone[28,29]. The exposed bone matrix is then degraded by CTSK and MMP9 and other secreted proteases[30,31]. Partial dephosphorylation of OPN by TRAP leads to dissociation of the ITGB3/OPN bond and frees the osteoclast to migrate to new areas of bone[32]. A redox active iron in TRAP catalyzes the generation of reactive oxygen species, including hydrogen peroxide, hydroxyl radicals and singlet oxygen, which facilitate bone resorption and degradation[33].

Many of these features of osteoclastogenesis outlined above were apparent in the PAM patient lung and Npt2b[−/−] mouse lung. CTSK, TRAP and CALCR expressing cells were found attached to the microliths, many of them multinucleated (Fig. 4j, n, s). The gene expression programs in the human PAM lung that were identified by RNA-seq demonstrated marked upregulation of osteoclast genes in monocyte/macrophages, including key transcription factors NFATc1, NFkB (IKBKB) and MITF, fusion proteins OCSTAMP and DCSTAMP, bone degrading proteins ATP6V0D2 and CLCN7, TRAP, CTSK and MMP9, and cell surface receptors colony stimulating factor-1 receptor (CSF1R) and CALCR. The upregulation of many of these genes was documented by RT-PCR in BAL cells from Npt2b[−/−] mice and in Npt2b[+/+] mice after adoptive transfer of microliths or synthetic hydroxyapatite microspheres, including those characteristic of pre-fusion osteoclasts such as *Acp5* and *Mmp9*, and also mature and activated osteoclast markers such as *Ctsk* and *Calcr*. Signature osteoclast enzymatic activities that were documented to be upregulated in the Npt2b[−/−] lung included CTSK (Fig. 4t, u) and TRAP (Fig. 4g–j), and BAL cells from hydroxyapatite challenged Npt2b[+/+] mice acquired the capacity to degrade bone (Fig. 5f–p). The delayed clearance of adoptively transferred microliths in CCR2[−/−] mice compared to control mice suggests that recruited monocytes play an important role in microlith clearance. Alveolar macrophages do not express CCR2, so reduced plasticity of AM limiting osteoclast differentiation is not a likely explanation for the clearance defect in this model[34]. Pseudotime analysis revealed a differentiation trajectory most consistent with monocytes as the primary cell of origin for the pulmonary osteoclast-like cells. However, osteoclast differentiation and microlith clearance were not inhibited completely in CCR2[−/−] mouse lung, indicating tissue resident AM also contribute to osteoclast-like MNGCs differentiation and stone degradation[35]. This conclusion is also supported by the expression of CD163 and CD68 in PAM_Mac2 and PAM_Mac3, which were the most osteoclast-like subtypes. BAL AM and bone marrow-derived monocytes (BMDM) where induced to become TRAP[+] MNGC with actin ring formation and bone pitting capacity when cultured with M-CSF and high dose (100 ng/ml) RANKL, including AM from CCR2[−/−] mice that are wholly derived from fetal progenitors, demonstrating that tissue resident AM are capable of becoming osteoclasts (Supplementary Fig. 6b). At lower doses of RANKL (25 ng/ml), however, osteoclast formation was only present in BAL AM that were isolated from mice pre-challenged with synthetic hydroxyapatite microspheres. The demonstration that anti-RANKL antibody slowed the clearance of microliths from the lung provides evidence that osteoclast-like MNGCs play a role in limiting microlith accumulation in the PAM lung. Although RANKL was not well represented in single-cell analysis of the

PAM lung (Supplementary Fig. 4), and RANKL was at the limit of detection in mouse BAL fluid due to dilution (not shown), AT2 cell lysates from Npt2b$^{-/-}$ mice contained almost 8-fold greater amounts of RANKL than Npt2b$^{+/+}$ AT2, suggesting that the alveolar epithelium is the source for this osteoclastogenic cytokine. It is also possible that T lymphocytes and NK cells represent a source for RANKL in the PAM lung, although differences in RANKL protein levels in BAL cell lysates from Npt2b$^{-/-}$ and Npt2b$^{+/+}$ mice were not significant (Supplementary Fig. 9).

Prior reports of true osteoclast-like cell differentiation in the lung have included descriptions of MNGCs within lung tumors, most likely as a result of DNA mutations and aberrant gene expression[36]. Inflammatory macrophages (IMs), MNGCs and foreign body giant cells (FBGCs) that express osteoclast markers are present in multiple benign lung diseases, however, including pneumoconiosis (e.g. silicosis and asbestosis) and granulomatous lung diseases (e.g. granulomatosis with polyangiitis, sarcoidosis and tuberculosis)[37–39]. Although there is no consensus on the cellular features and functions that reliably distinguish osteoclasts from IM, MNGC or FBGC, polarized expression of CTSK and ATP6V0D2 at the osteoclast/bone interface, formation of actin rings and a ruffled border, the expression of the CALCR in myeloid cells, and ability to degrade both the mineral and matrix components of bone are perhaps the most unique to osteoclasts, and CALCR expression, bone resorption, and actin ring formation were documented in our models (Figs. 5m, 4p–s, 5o, p). While we found that CALCR protein expression is upregulated in myeloid cells and TRAP$^+$ MNGC of Npt2b$^{-/-}$ mice and PAM patients (Fig. 4p, r, s), we also note that CALCR is expressed in pulmonary airway epithelium (Fig. 4o, q), and has been reported in other non-myeloid compartments in kidney, uterine corpus and cardiac myocytes[40].

Serum phosphate levels in Npt2b$^{+/+}$ mice and Npt2b$^{-/-}$ mice increased with increased dietary phosphate intake, as did BAL phosphate levels in Npt2b$^{-/-}$ mice, but BAL phosphate levels remained constant in Npt2b$^{+/+}$ mice despite 20-fold differences in the phosphate content of their diet and 2-fold differences in serum phosphate levels. These data reveal a previously unrecognized role for Npt2b in maintaining alveolar phosphate homeostasis.

We have previously reported that microlith accumulation can be limited or reversed by feeding mice a low-phosphate diet[11], and suggested LPD mediated upregulation of Slc20a1 (Pit1) and Slc20a2 (Pit2) gene expression on AT2 may have compensated for loss of Npt2b-mediated phosphate transport activity. The finding that LPD decreased phosphate and calcium levels in BALF is consistent with augmentation of calcium and phosphate export from the ELF. The Pit1 and Pit2 transporters are known to be upregulated by VitD3[41,42], which was elevated in the serum of LPD treated animals compared to RD treatment. TRPV6 and S100G mediated calcium transport is also known to be VitD3 dependent and gene expression of these transporters was upregulated in LPD compared to HPD. We submit that upregulation of alternative ion transporters may play a role in reducing calcium and phosphate levels in the ELF of LPD treated Npt2b$^{-/-}$ mice compared to RD mice, although definitive proof of the role of individual transporters will require the availability of specific inhibitors of phosphate transport or knockout animals.

Dietary phosphate intake is also known to regulate bone osteoclast activity[43,44]. When phosphate intake and serum levels drop, bone osteoclasts are activated to liberate phosphate and attenuate hypophosphatemia. In the Npt2b$^{-/-}$ model, in addition to modestly lowering serum and BAL phosphate concentrations, LPD increased expression of osteoclast signature genes *Acp5*, *Mmp9* and *Tnfrsf11a* (RANK) in BAL cells (Fig. 7c–e). Surprisingly, HPD treatment resulted in a marked increase in OPG in the alveolar lining fluid (Fig. 7f). OPG functions as a soluble decoy receptor that inhibits osteoclast differentiation by binding and neutralizing RANKL[23]. Elevated phosphate concentrations are known to inhibit osteoclast differentiation both by upregulating

OPG levels and by direct action on osteoclast precursor cells[45]. In HPD treated Npt2b$^{-/-}$ mice with elevated OPG in BAL, there were fewer TRAP$^{hi}$ MNGCs in the lungs compared to RD and LPD treated Npt2b$^{-/-}$ mice, which is similar to the findings in Npt2b$^{-/-}$ mice treated with neutralizing RANKL antibody. Collectively, these results indicate that the microlith burden in Npt2b$^{-/-}$ mice is modulated by dietary phosphate intake, in part through differential effects of OPG and RANKL on osteoclast differentiation and activity.

Analysis of BAL fluid and serum from the Npt2b$^{-/-}$ mouse previously identified monocyte chemoattractant protein-1 (MCP-1) and surfactant protein D (SP-D) as potential PAM biomarkers, which were then shown to be elevated in the serum of patients with PAM[11]. In adoptive transfer experiments, MCP-1 peaked at d7 post microlith challenge and returned to baseline at d28, coincident with clearance of both microliths and resolution of inflammation[11]. The abundance of OPG in the proteome of the human and murine microliths led us to the finding that the protein was also elevated in BAL fluid and serum of the Npt2b$^{-/-}$ mouse, and in the serum of PAM patients, positioning OPN as another potential PAM biomarker. OPN is a multifunctional protein which is highly expressed in bone, as well as at various levels in myeloid, smooth muscle, endothelial, and epithelial of other organs, secreted in response to cellular stress, including exposure to high-phosphate environments[46]. This widespread expression pattern precludes utility for OPN as a diagnostic biomarker, but it may have value as a prognostic or predictive biomarker of lung injury (i.e. microlith burden) or treatment response, as is being studied in idiopathic pulmonary fibrosis and other lung diseases[12–15].

It is tempting to speculate that the fibrosis that ultimately occurs in the PAM lung may be a consequence of collateral tissue injury from osteoclast products such as hydrochloric acid, TRAP, CTSK and MMP9. We intend to examine this question using animals that are deficient in genes that encode these phosphatases, proteases and ion pumps, and well as those that regulate osteoclastogenesis (e.g. RANKL, NFATC1). Osteoclasts may also contribute to fibrosis through mechanisms that are operative in the coupling of osteoclasts and osteoblasts in bone remodeling, including the release of IGF-1 and TGFβ from lung matrix, secretion of clastokines (S1P, Sema4D, C3a, etc), and direct cell-cell interactions (such as through Eph/EphR or reverse signaling through RANKL)[47]. These concepts are worth exploring since, if proven correct, there are many FDA approved therapies that target osteoclast viability and function which could potentially impact the development and progression of pulmonary fibrosis.

There are several limitations in the present study, most related to the rarity of PAM and limited availability of tissue samples. Multinucleated cells were likely excluded during scRNAseq processing and underrepresented in the analysis. The control lung tissue used in our studies included explants declined for organ donation, and RNA expression may have been altered by prior medical and physiologic conditions. Finally, although scRNAseq can suggest potential origins for pulmonary osteoclast-like cells, definitive conclusions will require careful lineage tracing, which we are now pursuing.

In summary, studying the mechanisms of microlith clearance in the rare lung disease PAM led us to identify an osteoclast-like pulmonary myeloid cell that plays an important role in limiting the stone burden in the lung. Augmenting the numbers or functioning of these cells represents a promising strategy for PAM therapeutic trials. Future studies from the laboratory will focus on whether pulmonary osteoclast-like cells play a role in the response to other particulate challenges and in development of pulmonary fibrosis in diseases such as silicosis and asbestosis.

## Methods

### Ethical approval

Human sample collection and laboratory analysis were approved by the Institutional Review Board at The University of Cincinnati College

of Medicine (#2013-8157) and informed consent was obtained per protocol.

## Mice

The epithelium-targeted Npt2b$^{-/-}$ mouse model was developed by breeding mice homozygous for floxed Slc34a2 with mice expressing Cre recombinase under the influence of the sonic hedgehog (Shh) promoter, as previously reported[11]. Female Npt2b$^{-/-}$ mice and their wild-type littermate controls (Npt2b$^{+/+}$), 5- to 6-week-old or 20- to 28-week-old were used in low and high phosphate diet treatment experiment, 8- to 10-week-old for all the other experiments. Male 20- to 28-week-old Npt2b$^{-/-}$ mice were used for microlith isolation. C57BL/6J mice and CCR2$^{-/-}$ mice, B6.129S4-Ccr2$^{tm1lfc}$/J were purchased from Jackson Laboratories (Bar Harbor, ME). Eight- to ten-week-old female CCR2$^{-/-}$ mice and wild-type C57BL/6J mice were used in all experiments. For terminal experiments, mice were euthanized by intraperitoneal injection of Euthasol (Virbac, Fort Worth, TX). All animals were maintained in a specific pathogen–free facility and were handled according to a University of Cincinnati Institutional Animal Care and Use Committee–approved protocol and National Institutes of Health guidelines.

## Preparation of crystals

Synthetic hydroxyapatite particles (10 μM average size) were purchased from Fluidinova (article 501203, Moreira da Maia, Portugal). Mouse microliths were isolated from the lungs of 20- to 28-week-old male Npt2b$^{-/-}$ mice after euthanasia by intratracheal instillation of 3 ml of dispase (50 caseinolytic units/ml, Corning, NY), submersion of the organ in 1 ml of dispase for 45 min at room temperature, and transfer to a culture dish containing water. The parenchymal lung tissue was gently teased from the bronchi, then homogenized. For some experiments, mouse microliths were isolated without dispase treatment. Human PAM microliths were isolated from the lung explant of a PAM infant undergoing lung transplant, by gently teasing the lung parenchyma apart with forceps in a culture dish containing water. Microliths were collected by centrifugation, and washed 5 times with cell culture grade water.

## Scanning electron microscopy and energy-dispersive X-ray spectroscopy analyses

Microliths were collected without dispase treatment from 20- to 28-week-old male Npt2b$^{-/-}$ mice and a PAM patient explant as described above. Microliths were incubated with distilled water or 0.02% SDS for 5 min, then washed 5 times with water to remove residual SDS and dried. The samples were placed on conductive carbon tape. Imaging was performed at 2 kV on a scanning electron microscope (FESEM, FEI Scios DualBeam, Germany). The elemental analyses were obtained at 15 kV beam voltage using an EDAX Octane Elite Super detector.

## Protein extraction and protein identification

Microliths were incubated in buffer containing 50 mM tetraethylammonium bicarbonate (TEAB), 5 mM Tris, and 75 mM EGTA with sonication and end over end shaking in the cold room. The supernatant was then buffer exchanged with 50 mM TEAB for 3 cycles using 3 kDa filters (Amicon). The final volume of the sample was brought to 100 μl with 50 mM TEAB. The samples were reduced with 20 μl of 50 mM tris-(2 carboxyethyl) phosphine (TCEP), alkylated with 10 μl of 200 mM methyl methanethiosulfonate (MMTS), digested with trypsin and dried in a lyophyllizer. Samples were analyzed by nanoLC-MS/MS on a Sciex 5600 quadrupole-TOF system and identified using mus musculus or the homo sapien databases with the Protein Pilot (ver 5.0, rev 4769) program (Sciex) as described previously[48].

## Lipid extraction and lipid identification

Total lipid extracts (TLEs) were collected using a modified Folch extraction[49,50]. Briefly, chloroform, methanol and water were added to the microliths to a final ratio of 8:4:3 respectively. The samples were vortexed for 30 sec, chilled on ice for 5 min, vortexed again and centrifuged at 10,000 × g for 10 min at 4 °C. The lower organic lipid containing layer was removed, dried in vacuo and reconstituted in 50 μl of methanol. Due to the differing yields of TLE from Npt2b$^{+/+}$ and Npt2b$^{-/-}$ mice, the BALF pellets were reconstituted in 100 μl and 1500 μl of methanol, respectively. Samples were analyzed using mass spectrometry, using LC-MS/MS parameters and identification algorithms outlined by Kyle et al.[51]. A Waters Aquity UPLC H class system interfaced with a Velos-ETD Orbitrap mass spectrometer was used for LC-ESI-MS/MS analyses. Reconstituted TLEs were injected onto a reversed phase Waters CSH column (3.0 mm × 150 mm × 1.7 μm particle size), and lipids were separated over a 34 min gradient (mobile phase A: ACN/H2O (40:60) containing 10 mM ammonium acetate; mobile phase B: ACN/IPA (10:90) containing 10 mM ammonium acetate) at a flow rate of 250 μl/min. Samples were analyzed in both positive and negative ionization modes using HCD (higher-energy collision dissociation) and CID (collision-induced dissociation). The LC-MS/MS raw data files were analyzed using LIQUID and all identifications were manually validated[51] which included examining the fragmentation spectra for diagnostic ions and fragment ions corresponding to the acyl chains, the precursor mass isotopic profile and mass ppm error, the extracted ion chromatograph, and retention time. To facilitate quantification of the lipids, a reference database for BALF lipids identified from 4 of the raw data files (2 WT and 2 KO) using LIQUID was created and features from each analysis were aligned to the reference database based on their identification, $m/z$ and retention time using MZmine 2[52]. A separate reference dataset was created from both of the microlith samples analyzed. Aligned features were manually verified and peak apex intensity values were exported for subsequent statistical analysis.

## Single-cell RNA-seq analysis

Cell Ranger R kit version 2.1.1 (10X Genomics) was used to process raw sequencing data and paired-end sequence alignment to the human genome (hg19). We used Sincera[53,54] and Seurat 2[55,56] for downstream analysis. For pro-filtering, we required that cells express more than 500 genes (transcript count > 0), and that less than 10% of transcript counts mapped to mitochondrial genes. Genes with transcripts detected in less than 2 cells were removed. A total of 14,210 cells and 21,075 genes passed the pro-filtering step and were used for further analysis. Transcript counts were normalized by dividing by the total number of transcripts in each cell multiplied by 10,000 and log-transformation (normalized count +1). For clustering, principal-component analysis was performed for dimension reduction. Reduced dimensions were used for cell cluster identification using the Jaccard-Louvain clustering algorithm[57]. For refined cell type mapping, we performed integrated PAM and control data analysis using Human Lung Cell Atlas[58] as a reference and SingleR[59] as the machine learning method. We applied MNN[60] and Seurat3 CCA[61] to minimize batch effect and optimize the data integration. We then extracted the immune cell cluster for subclustering using iterative Jaccard-Louvain clustering algorithm and identified five macrophage subclusters[57]. For this study, focused primarily on macrophage subtypes. All the downstream analysis including signature markers comparison, functional enrichment and pathway analysis were done in macrophages of PAM and donor samples. Differential expression of genes across the PAM lung sample and control lung sample were tested using a nonparametric binomial test[57]. Cell specific signature genes were defined based on following criteria: (1) <0.1 false discovery rate (FDR) of the binomial test, (2) minimum two-fold effective size, (3) detection in at least 20% of cells in the group. Monocle 3[62] was used to reconstruct a trajectory model for PAM monocytes and PAM macrophages.

## Histology and immunohistochemistry

Mouse and human lung tissues were fixed with 10% buffered formalin phosphate, embedded in paraffin, and stained with hematoxylin and eosin (H&E). The von Kossa technique was used for staining microliths by incubation of histological sections with 3% silver nitrate, exposure to UV light for 30 min, and counterstaining with Nuclear Fast Red (Newcomer Supply; Middleton, WI). For TRAP staining, tissue sections were incubated with pre-warmed (37 °C) TRAP staining mix (University of Rochester Medical Center TRAP protocol) for 1 h and counterstaining with Harris hematoxylin for 5 sec. Immunohistochemical analysis was performed on formalin-fixed, paraffin embedded material. Five micrometer sections were deparaffinized and rehydrated with dH$_2$O. The sections were subjected to antigen retrieval with citrate buffer, blocked with normal serum, and probed with specific primary antibodies anti-OPN (ab8448, polyclonal, 1:7500) and anti-cathepsin K (ab19027, polyclonal, 1:750) obtained from Abcam (Cambridge, UK), anti-CALCR LS (LS-A769, polyclonal, 9ug/ml) obtained from LS Bio (Seattle, WA) followed by horseradish peroxidase linked anti-rabbit secondary antibody (Cell Signaling Technology Inc., Beverly, MA). Slides were developed with DAB substrate, counter stained with hematoxylin, dehydrated, and mounted.

## Adoptive transfer of microliths

Microliths recovered from 20 to 24-week-old male Npt2b$^{-/-}$ mice lungs were washed with water 5 times then dried. Dried microliths were resuspended in normal saline at a concentration of 100 mg/ml, and 100 μL of the suspension was intratracheally instilled into each mouse, followed by histologic and radiographic analyses at time points indicated.

## BAL cell and fluid collection

C57BL/6J (8- to 10-week-old female) and CCR2$^{-/-}$ (8- to 10-week-old female) mice were sacrificed and subjected to BAL by intratracheal intubation followed by 5 cycles of instillation and aspiration of one ml of 0.9% saline. The lavage was pooled on ice and BAL cells were pelleted by centrifugation at 500 × g for 10 min at 4 °C. Gene expression in BAL cells was assessed by RT-PCR, and the supernatant from the first lavage was used for ELISA and colorimetric assays.

## Cell culture

Primary mouse BMDM or AM were prepared from 8 to 10-week-old female C57BL/6J mice. BMDM were isolated with a monocyte isolation kit (Miltenyi Biotec, Bergisch Gladbach, Germany) following the manufacturer's instructions. AM were collected from BALF as described above. BMDM and AM were cultured in Dulbecco's Modified Eagle's Medium (DMEM) + 10% heat-inactivated fetal bovine serum (FBS) supplemented with 20 ng/ml recombinant murine M-CSF (PeproTech, Rocky Hill, NJ) and 2 ng/ml recombinant murine RANKL (PeproTech). Medium was changed every 2 to 3 days. At day 3 or day 7 of culture, total RNA was purified from cells for RT-PCR analysis.

## Measurement of analytes

Phosphate and calcium levels in serum and BALF were measured using the Phosphate Colorimetric Assay Kit (BioVision, Milpitas, CA) and Calcium Colorimetric Assay Kit (BioVision), respectively. ELISAs were used to measure SP-D (Rat/Mouse SP-D kit, YAMASA, Tokyo, Japan), and MCP-1 (CCL2 /JE/MCP-1 ELISA kit, R&D Systems Inc., Minneapolis, MN) in mouse serum. Osteopontin in mouse BALF was measured using the Mouse/Rat Osteopontin DuoSet ELISA kit, and human serum osteopontin was measured using the Human Osteopontin DuoSet ELISA kit (R&D Systems Inc.). Phosphate homeostasis hormones were measured in mouse and human serum using the FGF-23 ELISA Kit (Kainos Laboratories Inc., Tokyo, Japan), the mouse or human PTH 1-84 ELISA Kit (Immutopics, San Clemente, CA), the 25-Hydroxy Vitamin Ds EIA kit (Immunodiagnostic Systems, Boldon, UK) and 1,25-Dihydroxy Vitamin D EIA kit (Immunodiagnostic Systems). RANKL in cell lysate of AT2 and BAL cells were measured using the Mouse RANKL DuoSet ELISA kit (R&D Systems Inc.).

## Preparation of RNA and RT-PCR

RNA from murine BAL pellets, primary AT2 cells or cultured BAL or AT2 cells was isolated using RNAzol RT (Molecular Research Center, Cincinnati, OH). RNA from tissues of Npt2b$^{-/-}$ and Npt2b$^{+/+}$ mice was isolated using the RNeasy Micro Kit (Qiagen, Hilden, Germany). cDNA was generated using the SuperScript III First-Strand Synthesis System (Life Technologies, Carlsbad, CA). After first-strand complementary DNA synthesis using SuperScript III Reverse Transcriptase (Invitrogen, Carlsbad, CA), quantitative RT-PCR was performed using a SYBR Green Master Mix (Applied Biosystems, Foster City, CA) and primer pairs outline in Supplementary Table 3, including β-actin as an internal control. Primers were obtained from Integrated DNA Technologies.

## Phosphate diets

Regular-phosphate content chow (RD), low-phosphate chow (LPD) and high-phosphate (HPD) chow containing, respectively, 0.7% /0.02% /2.0% phosphate, 16.3% /16% /16.3% protein, 66.3% /68.0% /63.3% carbohydrate, 5.0% /5.0% /5.0% fat, 1.2% /1.2% /1.2% calcium and vitamin D3 (2900 IU/Kg) were specially prepared by Harlan Laboratories (Madison, WI) and purchased from Harlan Sprague Dawley (Indianapolis, IN, USA). For some experiments, 0.1% rather than 0.02% phosphate containing chow was used for the LPD diet. 5- to 6-week-old female Npt2b$^{+/+}$ and Npt2b$^{-/-}$ mice were fed with RD for 2 weeks as pretreatment, followed by RD, LPD or HPD for the indicated time intervals. For RT-PCR studies, mice were assigned to a diet for 1 week, lungs were explanted and homogenized for RT-PCR. For histological analysis, mice were treated with RD, LPD or HPD for 2 weeks, lungs were fixed for H&E and von Kossa staining. To determine the effect of dietary phosphate restriction and excess on microlith burden in mice with mature PAM lesions, 20- to 28-week-old female Npt2b$^{+/+}$ and Npt2b$^{-/-}$ mice were fed with 0.1% phosphate LPD, 0.7% phosphate RD or 2.0% phosphate HPD for 8 weeks, followed by analyses including chest x-ray, chest CT and biochemical analyses.

## MicroCT, X-ray imaging and bone density

After animals were anesthetized with isoflurane, two-dimensional (2D) chest x-ray images (In Vivo Multispectral Imaging System FX, Bruker) were obtained, and microCT scans (Inveon, Siemens) with respiratory gating were performed. For microlith burden assessment, a commercial graphic processing software package (Amira, FEI Company, Hillsboro OR) was used to segment the lung from other soft tissues in CT images. Mean lung density and percentage of high-density lung volumes (>1.5 g/cc and >1.25 g/cc, calculated from the Hounsfield units in CT images) were calculated from the entire lung in MATLAB. Total mass of the lung was calculated by multiplying average density by the CT-derived lung volume. Stone burden was calculated as the product of lung volume and the increase in lung density over the average density of wild-type lungs.

## Isolation of AT2 cells

Mice were sacrificed by intraperitoneal injection of Euthasol, and lungs were perfused with 10 ml of sterile normal saline via the pulmonary artery. The airway was cannulated via tracheostomy with a 20-gauge metallic angiocatheter, and 3 ml of dispase (50 caseinolytic units/ml, Corning) was instilled, followed by 0.5 ml of 1% low-melt agarose (warmed to 45 °C). Lungs were rapidly cooled on ice for 2 min, submerged in 1 ml of dispase for 45 min at room temperature, and transferred to a culture dish containing deoxyribonuclease I (100 U/ml) (Worthington Biochemicals, Malvern, PA). The parenchymal lung tissue was gently teased from the bronchi, and homogenized. Cell suspensions were filtered, collected by centrifugation, and panned over

prewashed 100-mm tissue culture plates coated with CD45 (#553076, monoclonal, 1:100) and CD16/32 (#553142, monoclonal, 1:100) antibodies (BD Biosciences, San Jose, CA). After incubation for 60 min at 37 °C in a 5% $CO_2$ atmosphere to promote adherence of contaminating macrophages and fibroblasts, the AT2 were gently decanted from the plate, collected by centrifugation, and counted. For the Npt2b$^{-/-}$ animals, differential centrifugation was used to separate microliths from the cells. Cell viability determined with trypan blue staining was routinely >90%, and cell purity determined by SP-C staining ranged from 75 to 90%.

### Fluorescent reflectance imaging (FRI)

Two commercially available fluorescent imaging probes (PerkinElmer Inc., Hopkinton, MA) were used to evaluate osteoclast functions in the lung. The fluorescent bisphosphonate imaging agent, OsteoSense 750EX (Osteosense) was used to tag microliths in the lung. The cathepsin K cleavage activated fluorescent probe; Cat K 680 FAST probe (Cat K) were used to detect cathepsin K activity in the lung. Npt2b$^{+/+}$ and Npt2b$^{-/-}$ mice were shaved around the chest prior to imaging, and 1 nmol Cat K or 1 nmol Osteosense was administered intratracheally. The fluorescence signal emanating from the chest was monitored using an in vivo imaging system (IVIS Spectrum, PerkinElmer) at 18 h post injection for Cat K, and at days 1, 3, 5, 7, 10 and 12 post-injection for Osteosense. Quantification of fluorescence intensity was performed by evaluating the total radiant efficiency ([photons/sec]/[$\mu W/cm^2$]) of the signal within a region of interest (ROI). The ROI was defined by an area with a radius of 0.94 cm$^2$ that encompassed the lungs of the mice (Fig. 3s) and the total radiant efficiency of Osteosense was normalized to that present at day 3, to limit the mouse-to-mouse variation in delivery of the fluorescent probe.

### Bone resorption assay

The hydroxyapatite nanopowder, nanoXIM-HAp203 (particle size 10.0 ± 2.0 $\mu$m, Fluidinova, Moreira da Maia, Portugal) was resuspended in normal saline at a concentration of 10 mg/ml, and 100 $\mu$L of the suspension or 100 $\mu$l of normal saline were intratracheally instilled into C57BL/6J mice. BAL cells were collected 14 days after adoptive transfer of saline or hydroxyapatite. Freshly collected BAL cells were plated on bovine bone slices (Immunodiagnostic Systems, UK) in 96-well plates (5.0 × 10$^4$ cells/well) and incubated with α-minimum essential medium (α-MEM; Gibco) containing 10% FBS in the presence of various concentrations (0–100 ng/ml) of recombinant murine RANKL (PeproTech) with 1:40 dilution of CMG12-14 (murine M-CSF producing cell line[63]) conditioned medium (equivalent to 30 ng/ml of recombinant human M-CSF). Medium was changed on day 2, 4, and 5. Cells in 96-well plates were fixed in formalin and stained for TRAP activity after 6 days in culture. For actin ring staining, the cells on bone slices were fixed on day 6 in formalin and permeabilized in 1% Triton X-100, rinsed with PBS, and stained with AlexaFluor 488-phalloidin (Invitogen). For the bone pitting assay, cells on bone slices were removed by scrubbing the bone surface with toothbrush, and resorption pits were visualized by incubation with 20 $\mu$g/ml peroxidase-conjugated wheat germ agglutinin and stained with 3,3′-diaminobenzidine (Sigma-Aldrich, St. Louis, MO)[64].

### Anti-RANKL antibody treatment

Microliths in the Npt2b$^{-/-}$ mice lung were labeled by intratracheal administration of Osteosense as described above. 250 $\mu$g/mouse of anti-mouse RANKL mAb (clone IK22-5, Bio X Cell, West Lebanon, NH) or a Rat IgG2a isotype control IgG (clone 2A3, Bio X Cell) were administrated intraperitoneally into Npt2b$^{-/-}$ mice three times per week for 20 days or 28 days. For the IVIS study, mice were pretreated with anti-RANKL for 7 days before Osteosense injection. The treatment effect on microlith dissociation and osteoclastogenesis were monitored by quantifying the residual fluorescence signal using IVIS or histological analysis of the lung.

### Statistics and reproducibility

Data are presented as mean ± SD, as indicated, of at least three independent experiments or biological replicates. Statistical analyses were performed using GraphPad Prism 5. Significance between two groups was determined using Welch's t-test. The evaluation of stone burden changes was determined by a two-tailed Ratio paired t test. In experiments in which more than two groups were compared, a one-way analysis of variance (ANOVA) was used followed by Bonferroni's multiple comparisons test. Differences were considered significant at $p < 0.05$.

### Reporting summary

Further information on research design is available in the Nature Portfolio Reporting Summary linked to this article.

## Data availability

scRNAseq data from PAM and donor lungs has been deposited to GEO, accession number: GSE199329. The mass spectrometry proteomics data have been deposited to the ProteomeXchange Consortium via the PRIDE [1] partner repository with the dataset identifier PXD039280. Microlith and BALF lipidomic data has been deposited into the MassIVE database https://massive.ucsd.edu/ProteoSAFe/dataset.jsp?task=526a505bbd9347f0b7ddcecc2156d425 under accession number MSV000091302. Source data are provided with this paper.

## Code availability

Default packages used included Seurat 9 (version 2.0 and 3.0), Harmony and Monocle 3 for scRNAseq analysis, and all code is publicly available. For details, please see "Method" section.

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

## Acknowledgements

Mass Spectrometry data were collected in the UC Proteomics laboratory on an instrument funded in part by a National Institutes of Health (NIH) shared instrumentation grant (S10 RR027015) to K.D.G. We thank the patients family for donation of their child's tissue for this research. We also thank Dr. Atsushi Sasaki, (Department of Pathology, Saitama Medical University School of Medicine, Japan) for sample collection and preparation. This article was prepared while Charles Ansong was employed at Pacific Northwest National Laboratory (PNNL). The opinions expressed in this article are the author's own and do not reflect the view of the National Institutes of Health, the Department of Health and Human Services, or the United States government.

## Author contributions

Y.U. designed and performed experiments, analyzed data, and wrote the manuscript. Y.T. designed and performed experiments and analyzed data. N.M.N. designed experiments and analyzed data. L.B.P. performed mouse husbandry, genotyping, histological and pulmonary physiological experiments. H.W. and Y.H. performed ELISA assays. J.J.Y. and E.Z. assisted with the IVIS assay. J.G.N. and J.C.G. isolated mouse BMMs. E.J.K. assisted with collection of patient samples. W.D.H. and K.D.G. performed mass spectrometry and analyzed proteomic data. J.G. and J.C.W. performed quantitative CT analysis, K.A.W-B. prepared tissues for single-cell RNA-seq and assisted with pathological analyses. S.Z. and Y.X. analyzed single-cell RNA-seq data. J.E.K. and C.A. performed mass spectrometry and analyzed lipidomics data. S.L.T. assisted with data analysis. Y.I. and G.A. provided PAM patient samples. F.X.M developed the concept, designed experiments, analyzed data, and wrote the manuscript. All co-authors contributed to final editing of the manuscript. Y.U. and Y.T. contributed equally to this project. Y.X. and F.X.M. co-supervised the project, Y.X. for the single-cell analysis and FXM for the laboratory and animal experiments. This work was supported by R01HL127455 (F.X.M), R37AR046523 (S.L.T), R01DK111389 (S.L.T) and Department of Internal Medicine, University of Cincinnati. The lipidomic research was supported by UO1 HL148860, in part in the Environmental Molecular Sciences Laboratory, a DOE Office of Science User Facility sponsored by the Biological and Environmental Research program under Contract No. DE-AC05-76RL01830.

## Competing interests

The authors declare no competing interests.

## Additional information

[1]Department of Internal Medicine, Division of Pulmonary, Critical Care, and Sleep Medicine, University of Cincinnati College of Medicine, Cincinnati, OH, USA. [2]Neonatology, Perinatal and Pulmonary Biology, Cincinnati Children's Hospital Medical Center, Cincinnati, OH, USA. [3]Department of Cancer Biology, University of Cincinnati College of Medicine, Cincinnati, OH, USA. [4]Center for Pulmonary Imaging Research, Cincinnati Children's Hospital Medical Center, Cincinnati, OH, USA. [5]Division of Pathology and Laboratory Medicine, Cincinnati Children's Hospital Medical Center, Cincinnati, OH, USA. [6]Biological Sciences Division, Pacific Northwest National Laboratory, Richland, WA, USA. [7]Department of Pathology, Washington University School of Medicine, St. Louis, MO, USA. [8]Department of Diffuse Lung Diseases and Respiratory Failure, Clinical Research Center, National Hospital Organization Kinki-Chuo Chest Medical Center, Osaka, Japan. [9]Department of Chest Diseases, Faculty of Medicine, Pamukkale University, Pamukkale, Turkey. [10]Departments of Pediatrics and Biomedical Informatics, University of Cincinnati School of Medicine, Cincinnati, OH, USA. ✉e-mail: ueya@sapmed.ac.jp; yan.xu@cchmc.org; frank.mccormack@uc.edu

