## [Peer Review File · Nature Communications]

Insights into pulmonary phosphate homeostasis and osteoclastogenesis emerge from the study of pulmonary alveolar microlithiasisREVIEWER COMMENTS

Reviewer #1 (Remarks to the Author):

The authors of "Novel insights into pulmonary phosphate homeostasis and osteoclastogenesis emerge from the study of pulmonary alveolar microlithiasis" impressively demonstrate a role for osteoclast like cells in microlithiasis. this is a very nice, novel (among >1200 pubmed entries with microlithiasis, 0 came back including osteoclast or osteoclastogenesis) and interesting manuscript. however i have a few remarks and comments.

major comments:

- the role of RANKL in the lungs is not well represented in the paper. where does the RANKL in vivo actually come from? the authors should address this question, where applicable.

- additionally an therapeutic approach with additional RANKL in the system either by intratracheal application or systemically would be informative. additional osteoclastogenesis, in particular locally, would be beneficial in microlithiasis.

- is it possible to stain these microliths with TRAP to detect osteoclasts directly o the surface of these structures? It would be certainly interesting to directly investigate these cells, since huge, multinucleated OCs or OC-like MGCs are not part of the single cell transcriptomics, due to size.

- concerning the scRNAseq, it is obvious that osteoclastogenic gene signature is dominant in the PAM monocytes/macrophages. however in my opinion the single cell analyses should be broadened to cover also other cells within the PAM cells that are not mentioned at all. there are certainly more subpopulation present the BAL of this patient. for instance a group of cells low in CD115 and negative for TRAP/ACP5, DC STAMP or ATP6VOD2 (see Fig. 2).

are there osteogenic cells present?

what is the origin of the osteoclast like cells in the lung. analysis of scRNAseq shows that Mph in PAM lungs (blue)are different from healthy macrophages (Mph) (orange). as is seems, there are no regular alveolar Mph left in the PAM patient. Are the Mphs in the PAM patient all derived from alveolar Mph? Are they recruited to the lung? Staining of PAM lungs with markers of alveolar Mph such as CD163 would be informative.

another question remaining is, why OCSTAMP and MMP9 are underrepresented. recent publications show a trajectory of osteoclastogenesis (DOI: 10.1038/s42255-020-00318-y). The authors could apply some the analysis to their scRNAseq. It would be interesting to see whether there are different subpopulations present (monocytes, macrophages, pre-OCs or pre-MGCs) that aquire the genesignature of MGCs and/or OCs.

- it is unclear to me why the authors included and analyzed osteopontin. this is clearly distracting. osteopontin expression has been shown to be increased in different compartments in various pathological conditions such as multiple myeloma (DOI: 10.1186/1756-8722-4-22), lung cancer (DOI:<https://doi.org/10.1097/JTO.0b013e3181ae2844>), blood eosinophilia (DOI:<https://doi.org/10.1016/j.jaci.2012.06.010>) just to name a few. It is therefore difficult to use as a biomarker.

- the experiments using different amounts of dietary phosphate show that high phosphate reduces the amount of OC like cells in the lung. also, levels of OPG are higher in this condition. what about RANKL levels? like above i suggest to measure RANKL under these condicions.

the other thing is, that the cells that might produce the micoliths (presumably osteoblast like cells?) are not mentioned in the manuscript. i am aware that this is not the focus of the manuscript, but presumably also RANKL and OPG are produced by these bone forming cells, so it might be worthwhile to briefly analyze them. the experiment with the different phosphate levels in the diet suggest, that also their activity is modulated in the model. are they present in the single cell RNA data set?

- the experiment whether RANKL inhibition blocks microlith clearance is not well designed, as it uses

the highly pathological Npt2b^{-/-} mice, in which, as mentioned above, also the activity of the osteogenic cells varies. The results there are also not very striking in terms of effect size. In my view, using anti RANKL treatment in wt mice in the inducible model with intratracheal instillation is better suited to analyze the effect of the OCs like cells in the resolution of microlithiasis.

Reviewer #2 (Remarks to the Author):

The authors identified osteoclast genes and proteins within the lungs and extracted microliths of PAM patients and PAM mouse model Npt2b^{-/-}, suggesting a potential role for osteoclasts in regulation of these CaP microliths. Implantation of microliths (adoptive transfer) into wild type mouse lungs induced osteoclast-like cells. Dietary phosphate regulated osteoclast-like cell formation in lungs, with low phosphate diet increasing osteoclast-like cells and ultimately reducing microliths in Npt2b^{-/-} mice. Inhibition of RANKL signaling via neutralizing antibody prevented the formation of TRAP positive osteoclast-like cells and prevented the clearance of microliths. Altogether, these results demonstrate a plasticity of lung myeloid/macrophage cells to exhibit an osteoclast-like phenotype in response to microlith accumulation. These findings have significant implications for treatment of PAM patients and provide novel therapeutic strategies. In addition, these results push forward our understanding of phosphate regulation of osteoclast phenotypes beyond the bone tissue. Overall, these are fascinating data and significant findings. Below I have noted specific instances in which the authors should provide additional information, alter methodology, or modify their interpretation.

It is not clear what lineage markers the authors used to define the macrophage population in the single cell analysis. This information should be provided to better understand how the population was selected. Related to this question, were these markers specific to macrophages, or did they also include myeloid progenitor or dendritic cell lineages? Given that these lineages can all contribute to osteoclast differentiation in bone, it would be of interest to assess these populations as well (depending on whether they were already included in the original lineage selection). Another related question is what happens to the macrophage marker F4/80 (ADGRE). Osteoclasts in bone are negative for this gene (while mononucleated osteomacs exhibit F4/80 and TRAP activity). Presence of F4/80 within this population would be valuable in assessing whether expression is reduced as these cells take on an osteoclast phenotype, similarly to the noted decrease in EMR1 expression in PAM myeloid cells. Also, given the results in Figure 6, did the gene for RANK (TNFRSF11A) show up in this analysis?

Regarding the differential gene expression analysis of the single cell data, the methods state that the binomial nonparametric test was used, and differential expressed genes are FDR<0.1, effect size>2, and frequency>20%. However in the figure legend this is only listed for the results in panel 2d. It is not clear whether the osteoclast genes are significantly increased from the figure and legend. Full differential gene expression results should be provided as supplemental data.

In Figure 3g-r, it appears that the TRAP and CTSK positive cells in PAM lungs are multinucleated. However, the CALCR positive cells appear mononucleated. This raises the question of whether the CALCR positive cells are a separate population of myeloid cells. If possible, co-staining would be useful to determine if these signals colocalize, especially since the authors note CALCR expression as one of the defining osteoclast features in the discussion. The authors also mention in this section that multinucleated giant cells colocalized with microliths and were found adherent to microliths. From these images, it is not clear whether microliths are present. If so, it would be beneficial to indicate these with arrows. Interpretation of immunostaining in Figure 3g-r would also be aided by labeling the figure directly to show which proteins are being identified in which panels.

The authors cultured BAL cells following intratracheal challenge with hydroxyapatite spheres on bovine bone chips to assess resorptive activity. It is unclear why the authors used hydroxyapatite in this assay instead of adoptive transfer of microlith or Npt2b^{-/-} mice, which are what they showed had increased osteoclast gene expression. Given the additional components within the microliths, it would be preferable to at least include adoptive transfer of microliths or Npt2b^{-/-} mouse BAL cells as another

group in this experiment. Alternatively, the authors should show that similarly to *Npt2b*^{-/-} mice (Figure 3a-e) or adoptive transfer of microliths (Figure 4a-d), the wild type hydroxyapatite recipient BAL cells express osteoclast markers.

The increase in CTX1 in the media is promising (Figure 3u), but does not confirm osteoclast resorption; this may simply represent the presence of increased protease activity. The authors should assess whether these BAL derived cells are forming actin rings that are crucial for formation of a resorption lacunae. Alternatively, the authors could stain the bone chips with toluidine blue to identify resorption pits following the culture period.

Previous studies have shown that *CCR2*^{-/-} bone marrow cultures have reduced osteoclast differentiation (PMID: 26205994). Thus, is it possible that the AM population has reduced plasticity towards osteoclast differentiation in *CCR2*^{-/-} mice? Do AM cells express *CCR2*?

Osteoclast differentiation of bone marrow derived monocytes typically occurs in 4 days, with maximal osteoclast gene expression at day 3. After forming osteoclasts, these cultures don't typically last longer than 5 days, as osteoclasts begin to undergo apoptosis. Given the sustained gene expression with microliths or hydroxyapatite, I'd be interested to see if these cells still look like osteoclasts and how much cell death has occurred. Overall, these results would also be more informative if TRAP staining images of the differentiation cultures at Days 3 and 7 were included. Including images of multinucleated AM-derived osteoclasts would also be informative as to whether this cell population is able to differentiate into osteoclasts on their own, or whether they require recruitment of circulating monocytes. I would not necessarily conclude that CaP crystals potentiate osteoclast differentiation, unless there is a clear increase in TRAP positive multinucleated osteoclasts. If no quantification of TRAP staining can be performed, I would restate the sentence (lines 216-218) to specifically indicate osteoclast gene expression.

The concept of targeting osteoclasts to prevent fibrosis in PAM lungs is very interesting, and is somewhat analogous to the treatment of fibrous dysplasia patients with anti-RANKL therapy. It is possible that rather than being mediated by protease-induced damage, the fibrosis may also result from osteoclast-derived factors that contribute to tissue regeneration (termed coupling in bone remodeling).

Reviewer #3 (Remarks to the Author):

A review of the manuscript from Uehara et al., "Novel insights into pulmonary phosphate homeostasis and osteoclastogenesis emerge from the study of pulmonary alveolar microlithiasis". PAM is a rare autosomal recessive lung disease in which microliths accumulate in the alveoli and is caused by mutations in the *SLC34A2* gene which encodes a sodium-phosphate co-transporter. In this study, the authors expand on current data in terms of phosphate biology. In the pathogenesis of PAM a they identify an osteoclast population/program derived from monocyte/macrophages in the PAM lung using a mouse (*Npt2b*^{-/-}) model and samples from an individual with PAM. They identify that the microliths induce osteoclast-like activity when transferred into healthy mice. Low phosphate diet increased the osteoclast-pattern of gene expression and was associated with increase clearance of microliths. A role for RANKL was identified in this process. High dietary phosphate led to elevated levels of osteoprotegerin (a decoy receptor from RANKL) and led to increased microlith burden. This study highlights the inability of the alveolar space to regulate phosphate metabolism in PAM and the identification of an osteoclast program opens new therapeutic research avenues. Multiple experiments are performed and summarized briefly here. First, a detailed analysis of PAM microliths (mice and one PAM patient) was performed and identified many proteins including osteoclast proteins. Osteopontin was identified as significantly elevated in PAM mice and patients (n=14). While statistically significant, the levels are variable in PAM patients. ScRNA-Seq of lung cells from a 2 year old PAM patient is performed vs a normal control sample (n=1 of each) and identified an osteoclast signature in PAM macrophages. Some of these genes were then validated in BAL cells. Multinucleated giant cells were identified, frequently adjacent to microliths, in PAM lungs and

expressed TRAP, CTSK and CALCR. Enzymatically active osteoclast protease activity (CTSK) was demonstrated in BAL cells from Npt2b^{-/-} mice using an in vivo imaging system. Adoptive transfer of microliths into WT lungs was performed and led to increased expression of osteoclast genes. CCR2^{-/-} mice were used to determine the origin of cells with the osteoclast program and suggested monocytes are the major source and alveolar macrophages a minor source. These findings are further validated in an in vitro experiment culturing bone marrow derived monocytes of alveolar macs in osteoclast conditions with microliths suggesting the microliths potentiate the osteoclast program independent of mutations that cause PAM. Next the authors demonstrate that alterations in dietary phosphate concentration leads to changes in transport levels and phosphate/calcium levels in the alveolar space. High phosphate diet increase BAL phosphate and microliths however TRAP positive cells decreased. Phosphate transporter gene expression was decreased in AEC2 compared to low phosphate diet. In vivo imaging and RANKL neutralizing antibodies were used to demonstrate the role of RANKL in osteoclast formation using and assessing microlith clearance.

Although PAM is rare genetic disease the insights learnt by studying this disease would likely prove beneficial in understanding the pathobiology of other pulmonary diseases. Overall, this is a very well designed and rigorously conducted study that combines transcriptomic analyses, human samples, mouse genetic models, and pharmacologic therapies. In my opinion, the work by Uehara and colleagues meets Nature Communications high publication standards and therefore recommend to accept with minor revisions.

Major:

1) Fig. 2: the scRNA-Seq of a single human sample is key to the papers findings. Further data analysis of this scRNA-Seq would add to the paper. The authors focus on the differences in gene expression in lung macrophage populations between a patient and a control explant. Since scRNA-seq analysis of whole lungs was performed, it would be of interest to the reader to first show a combined UMAP of all cells in the patient and control explants and to annotate all cell clusters. It is also unclear to this reader whether the osteoclast-related genes were among the top differentially upregulated genes. Including a heatmap of the top differentially genes in patient macrophages vs control would be useful. Furthermore:

-although only n=1, are there differences in the AEC2 in the PAM patient (inflammatory, pro-fibrotic pathways?)

-what is the expression of the transporters described later in the human samples

-were MNGCs included in the scRNA seq or would these be excluded by the sample preparation of bioinformatic analysis

There appear to be two populations deemed macrophages in the scRNA seq of human lungs, perhaps more in the PAM patient. This heterogeneity is not addressed. Can the authors clarify the nature of these clusters and provide data if needed demonstrating the differences. The expression of canonical macrophage/myeloid markers in a supplemental figure would be useful in understanding this potential heterogeneity and its relevance to the osteoclast gene expression program.

Are there "normal" macrophage in the PAM scRNA-seq data?

Are there are subset of cells that are most osteoclast-like

Louvain clustering (or some alternative) applied to the myeloid clusters and violin plots of the osteoclast genes shown in 2b and c would provide more granular view of their expression.

Within the PAM myeloid cells, is there any evidence to suggest differentiation trajectories from AMs/monocytes to osteoclasts.

2. This is a rare disease thus understandable that scRNA-Seq was only feasible on one patient. However, if key findings could be validated, e.g. paraffin fixed sections, in other patients this would increase confidence in the human findings.

Minor:

-The link to osteopontin biomarker studies is clear from the figures but not the text.

-The conclusion that osteopontin is expressed in AEC1 and AEC2 cells is not well supported by the

immunostaining which would be improved by co-staining with canonical markers and higher resolution images (Fig 1o). Additional patient samples, if available, would strengthen this conclusion.

-Details of PAM patient mutation

-Line 143-144: The term infant is used for ages 0-1 years; consider using an alternative term.

-Line 144: "A heatmap illustrates upregulation of expression of key osteoclast related genes in PAM lung macrophages (Fig.2b-e)" should be revised to more accurately describe the different plots used in 2b-e

- Fig 1n: significant variation in serum osteopontin. Do the two outliers have worse disease, and are they the outliers in Fig5 g-i

-Line 143-144: The term infant is used for ages 0-1 years; consider using an alternative term.

Reviewer #4 (Remarks to the Author):

This manuscript is an overall well-written investigation of a rare lung disease, pulmonary alveolar microlithiasis (PAM), and the role of Npt2b in pulmonary phosphate homeostasis and microlith clearance. This work leverages scRNA-seq technology in an explanted diseased lung as well as murine models, including dietary interventions, to argue for a critical role for Npt2b and osteoclast-like pulmonary cells in phosphate homeostasis, which may support future discovery of therapeutics for this disease. While interpretations of the analysis are certainly limited by sample number, this would be the first published scRNA-seq analysis of PAM lung, and will provide important data for the scientific community as this is a rare disease with limited availability of patient tissue. The work could be improved by further details and attention to a few particular areas:

1. Osteopontin is suggested as a potential biomarker based on its elevated serum levels and murine BALF levels. Additionally, IHC staining suggested it localized to AECI, AECII, and microlith adjacent myeloid cells in the PAM lung. How does this expression correlate with SPP1 (osteopontin) expression in the scRNA-seq data? I.e which cell types are predominantly expressing this and does that change from control to disease? For instance, scRNA-seq data in multiple ILDs has demonstrated a distinct population of macrophages with high expression of SPP1 that are expanded in fibrosis compared to controls, and suggest that macrophages, not alveolar epithelial cells, are the predominant source of osteopontin in fibrosis.
2. Additional data should be provided on the scRNA-seq analysis methods including how the lung samples were digested, 10X Genomics chemistry used, specific method for combining the samples, batch correction. Please include UMAPs of the lung sample overall clusterings identifying all cell types, as well as demonstration of gene expression for how the macrophages were identified. Differential expression testing results for the "upregulated key osteoclast related genes", or all genes between the PAM and control macrophages needs to be included rather than visualizations alone. Currently how the depicted genes were chosen and the level of significance for them is unclear.
3. In Figure 2E, a DotPlot may be a more informative by showing both the percentage of cells expressing each gene and the expression level across both the disease and control, rather than the percent expressing alone.
4. Are specific subgroups of macrophages able to be identified to suggest a more precise origin for the osteoclast like cells? Downregulation of a single gene, EMR1, is insufficient to suggest that there is osteoclast differentiation in AM of the PAM lung.
5. In Figure 5, the authors show compelling effects of dietary phosphate on calcific densities and stone burden in the lung. Do the authors have any further phenotyping of these mice such as physiologic lung function to know if this difference in stone burden translated to a meaningful physiologic effect? Were there any significant changes in weight, mortality, and/or other indicators of murine health with the dietary intervention?
6. In Figure 5 panels d-e change in stone burden is shown only for the 0.7% and 0.1% Pi diet. This information should be included for the 2.0% Pi diet as well.
7. In Figure 3 panels g-r, visual interpretation would be easier by including the stains (TRAP, CTSK, CALCR) on the left y axes, similar to the tissue type labeling above the figure.
8. There is no discussion of limitations of the work. This should be thoughtfully considered and

included within the discussion section.

9. Manuscript does not include information on data availability, where sequencing data will be deposited, etc.

REVIEWER COMMENTS

We thank the reviewers for their thoughtful critiques. We have expanded the single cell analysis, performed additional laboratory experiments, deleted original figures 2d,e, 4f-m and supplementary figure 2 and added multiple figures including Figs. 2a, 3, 4e,f,s, 5f-o, 6f, 7g, Supplementary figures 1,2,3,4,5,6,7,9

Reviewer #1 (Remarks to the Author):

The authors of "Novel insights into pulmonary phosphate homeostasis and osteoclastogenesis emerge from the study of pulmonary alveolar microlithiasis" impressively demonstrate a role for osteoclast like cells in microlithiasis. this is a very nice, novel (among >1200 pubmed entries with microlithiasis, 0 came back including osteoclast or osteoclastogenesis) and interesting manuscript. however i have a few remarks and comments.

Thank you for positive comments.

major comments:

1a. - the role of RANKL in the lungs is not well represented in the paper. where does the RANKL in vivo actually come from? the authors should address this question, where applicable.

RANKL was not detected in the transcriptomic analysis, other than a tiny signal in T cells and NK cells (**Supplementary Figure 4**). RANKL protein was 8 fold more abundant in cell lysates of isolated alveolar type II cells from Npt2b^{-/-} mice than Npt2b^{+/+} mice (**Fig. 7g**) and there was no between group difference in RANKL in lysates from BAL cells (**Supplementary Figure 9**), which contain lymphocytes and AM. In PAM, we believe the RANKL is primarily produced by alveolar type II cells.

1b. - additionally an therapeutic approach with additional RANKL in the system either by intratracheal application or systemically would be informative. additional osteoclastogenesis, in particular locally, would be beneficial in microlithiasis.

Excellent suggestion; this is a worthwhile experiment with great translational appeal. Unfortunately, recombinant mouse RANKL costs \$5-\$40 per µg (cheapest if purchased in bulk at \$5200/mg), and typical doses administered to animals by the i.p route in the literature are ~ 200-300 µg/mouse (PMID 31244859). There is no literature we can find for intratracheal dosing. Multiple doses would likely be required, resulting in costs of several thousand dollars per mouse at a minimum. When time and budgets allow, we intend to make our own recombinant RANKL and perform this experiment.

1c. - is it possible to stain these microliths with TRAP to detect osteoclasts directly on the surface of these structures? It would be certainly interesting to directly investigate these cells, since huge, multinucleated OCs or OC-like MGCs are not part of the single cell transcriptomics, due to size.

Fig. 4j and 4n depict a TRAP and CTSK positive multinucleated GC on the surface of a microlith. **In Fig. 4s**, a dual TRAP and CALCR stained osteoclast is shown enveloping a stone. In the inset, TRAP and CALCR macrophages are seen in what appears to be various states of fusion.

1d. - concerning the scRNAseq, it is obvious that osteoclastogenic gene signature is dominant in the PAM monocytes/macrophages. however in my opinion the single cell analyses should be

broadened to cover also other cells within the PAM cells that are not mentioned at all. there are certainly more subpopulation present the BAL of this patient. for instance a group of cells low in CD115 and negative for TRAP/ACP5, DC STAMP or ATP6VOD2 (see Fig. 2).

A more detailed single cell analysis was requested by all reviewers (see new **Figs. 2,3, S1, S2, S3, S4**). We performed an integrated PAM and control data analysis. Using Human Lung cell Atlas (PMID: 33208946) as a reference and SingleR (PMC6340744) as a machine learning method, we identified total of 23 cell types including 5 epithelial cell types (AT1, AT2, club/basal, ciliated, pulmonary neuroendocrine cells), 6 endothelial subtypes (cap1, cap2, artery, venous, bronchial vessel, and lymphatic endothelial cells), 2 mesenchymal cell types (matrix fibroblast and smooth muscle cells) and 10 immune sub-types (macrophage, proliferating macrophage, classical monocytes, nonclassical monocytes, dendritic cells, natural killer, T cells, B cells, mast, plasma). **Supplementary Figures 1-3** are added to show integrated macrophage clusters in a UMAP plot and expression of cell type specific makers in a dot plot.

We also extracted out the macrophage cells from PAM and control donor, and re-performed the subclustering analysis (**Fig. 3c-d**). We identified 5 macrophage subtypes from PAM and two macrophage subtypes from control donor. PAM-Mac 3 express multiple osteoclast markers (ACP5, DCSTAMP, RANK, MMP9, ATP6VOD2), and represent the most 'osteoclast-like' cells. PAM-Mac 4 is low in CD115 and negative for ACP5/DCSTAMP/ATP6VOD2. PAM-Mac4 expresses classical monocyte markers such as S100A8 and CD14 and shares other transcription similarity with monocytes. We speculate that this population represents the fusion of monocyte/macrophage precursors. **Supplementary Figures 1a-1c** are added to show the macrophage subclustering analysis, the expression of macrophage subtype markers and the expression of osteoclast markers across macrophage subtypes in dot plots. OCSTAMP and MMP9 are both expressed at low levels.

1e. are there osteogenic cells present?

We find no evidence of osteogenic cells in the PAM lung, based on negative expression of osterix in scRNAseq of PAM lung, an essential osteoblast differentiation factor (not shown), and negative IHC for osteocalcin (not shown) --arguably the most specific marker for osteoblasts. Although Runx2 and Atf4 were both expressed at higher levels than control, we did not detect osterix in the scRNAseq of PAM lung. We believe that microliths form in the alveolus much like renal stones form in the renal tubule, by spontaneous nucleation and growth in the presence of supersaturated calcium and phosphate concentrations in luminal fluid.

1f. what is the origin of the osteoclast like cells in the lung. analysis of scRNAseq shows that Mph in PAM lungs (blue)are different from healthy macrophages (Mph) (orange). as is seems, there are no regular alveolar Mph left in the PAM patient. Are the Mphs in the PAM patient all derived from alveolar Mph? Are they recruited to the lung? Staining of PAM lungs with markers of alveolar Mph such as CD163 would be informative

Additional subclustering of macrophage populations was performed (**Fig. 3 and Supplementary Figs. 2,3**) and identified 2 subpopulations in controls and 5 subpopulations in PAM patients, of which PAM_Mac2 and PAM_Mac3 were the most osteoclast-like based on gene expression profiles. Pseudotime analysis suggested that monocytes are likely the primary cell of origin for pulmonary osteoclasts, though there was also a trajectory consistent with a contribution from AM. Delayed degradation of adoptively transferred microliths in the CCR2^{-/-} mouse (that cannot recruit circulating monocytes) indicates that recruited monocytes are required for optimal clearance (**Fig. 5e**). The fact that cultured BAL cells from CCR2^{-/-} mice can form osteoclasts when incubated with high dose RANKL +MCSF indicates that tissue resident

AM are capable of osteoclastic differentiation (**Supplementary Fig. 6b**). Although macrophages share many markers with monocyte and dendritic cell lineages such as SPI1, HLA-DRB1 and CD4, the combination of CD163, CD68, MARCO, MSR1 identify a macrophage population distinct from other myeloid cells that may be contributing to osteoclast generation (**Supplementary Fig. 2**).

1g. . another question remaining is, why OCSTAMP and MMP9 are underrepresented. recent publications show a trajectory of osteoclastogenesis (DOI: 10.1038/s42255-020-00318-y). The authors could apply some the analysis to their scRNAseq. It would be interesting to see whether there are different subpopulations present (monocytes, macrophages, pre-OCs or pre-MGCs) that acquire the genesignature of MGCs and/or OCs.

As reviewer 1 pointed out, OCSTAMP and MMP9 were only detected in a subset of macrophage cells. Macrophage subclustering analysis suggested OCSTAMP is selectively expressed in PAM-Mac2 and 3; and MMP9 is selectively expressed in PAM-Mac3. Since we only have n=1 PAM lung for scRNAseq, we are not sure whether this observation is representative of PAM macrophages in general. In the revision, these data are included in **Figs. 2c and 3d**.

Although the expression of MMP9 and OCSTAMP are low, they are uniquely detected in PAM lung but not in control lung (**Supplementary Figure 3**). The figure at right is provided for references, but was not included in the manuscript.

UMAP visualization of 7,781 macrophage cells (6,643 PAM, 1,831 control) are shown. Cells are colored by condition (top); Feature plot demonstrating the expressions of MMP9 (middle) and OCSTAMP (bottom) on macrophage cells. B) Violin plot shows increase of MMP9 and OCSTAMP expression in PAM samples compare with control

1h. it is unclear to me why the authors included and analyzed osteopontin. this is clearly distracting. osteopontin expression has been shown to be increased in different compartments in various pathological conditions such as multiple myeloma (DOI: 10.1186/1756-8722-4-22), lung cancer (DOI:https://doi.org/10.1097/JTO.0b013e3181ae2844), blood eosinophilia (DOI:https://doi.org/10.1016/j.jaci.2012.06.010) just to name a few. It is therefore difficult to use as a biomarker.

We agree that because it is widely expressed and increased in a number of conditions, osteopontin would be a poor choice as a diagnostic biomarker, but it has potential utility as a predictive or prognostic biomarker of lung injury (i.e. microlith burden) or treatment response, in much the same way as a sed rate is useful to gauge disease activity in temporal arteritis or serum surfactant proteins A (SP-A) and D (SP-D) can be used to determine the degree of barrier dysfunction in states of lung injury (PMID: 10588595). OPN is studied as a promising biomarker in other chronic lung diseases, including idiopathic pulmonary fibrosis (PMID: 34675050, 32450241, 32104688, 16128620). In addition, osteopontin is found in microliths and is a ligand for osteoclast ITGB3 binding. Two of four reviewers were interested in OPN, so we have opted to retain it, but have tempered some of the language in results and discussion section.

1i. the experiments using different amounts of dietary phosphate show that high phosphate reduces the amount of OC like cells in the lung. also, levels of OPG are higher in this condition. what about RANKL levels? like above i suggest to measure RANKL under these conditions.

RANKL levels in BAL were at the lower limit of detection which is not surprising given the >100 fold dilution inherent in the technique. (GM-CSF is similarly difficult to detect in the alveolar lining fluid by BAL sampling, but clearly plays a pivotal role in the differentiation of AM). We are also not certain that the ELISA detects RANKL complexed with OPG, which is abundant in the BAL of mice treated with higher phosphate containing diets (**Fig. 7f**). As shown in the figure at right, there were a few animals with higher BAL RANKL levels on low phosphate diet compared to regular and high phosphate diet (below) in both $Npt2b^{-/-}$ (KO) and $Npt2b^{+/+}$ (WT) mice, but the results were not significant. This figure was not included, but a note was made in the text.

1j. the other thing is, that the cells that might produce the microliths (presumably osteoblast like cells?) are not mentioned in the manuscript. i am aware that this is not the focus of the manuscript, but presumably also RANKL and OPG are produced by these bone forming cells, so it might be worthwhile to briefly analyze them. the experiment with the different phosphate levels in the diet suggest, that also their activity is modulated in the model. are they present in the single cell RNA data set?

Please answer to 1e.

1k. the experiment whether RANKL inhibition blocks microlith clearance is not well designed, as it uses the highly pathological $Npt2b^{-/-}$ mice, in which, as mentioned above, also the activity of the osteogenic cells varies. The results there are also not very striking in terms of effect size. In my view, using anti RANKL treatment in wt mice in the inducible model with intratracheal instillation is better suited to analyze the effect of the OCs like cells in the resolution of microlithiasis.

We find no evidence that osteogenic cells play a role in the formation of microliths in the $Npt2b^{-/-}$ mice (see response to 1e above), we believe that nucleation and stone growth occurs spontaneously in Ca/PO_4 supersaturated alveolar lining fluid. So we do not think that osteogenic cells confound the interpretation of the experiment in **Fig. 7j,k**.

The model presented in **Fig. 7j,k** demonstrates delayed decay of the osteosense signal on microliths with anti-RANKL vs. control IgG treatment, consistent with delayed clearance of microliths by osteoclasts. The argument is supported by the data in **Fig. 7h,i**, showing that MNGC formation is decreased by anti-RANKL treatment. Collectively, we believe these experiments demonstrate a role for osteoclasts in the clearance of endogenous microliths.

Nonetheless, we attempted the adoptive transfer experiment that the reviewer requested multiple times using synthetic hydroxyapatite microspheres labeled with fluorophores linked with active vs. inactive zoledronate (5FAM-Zol and AF647-Zol, Biovinc), but had trouble obtaining a stable baseline IVIS signal after initial instillation. This could be due to decay in the fluorescent signal over the course of the week long experiment, or possibly because a portion of the fluorescent particles continue to migrate ventrally over time (with a confounding influence on the fluorescent signal measured over the chest) in the quadrupedal animal .

Reviewer #2 (Remarks to the Author):

The authors identified osteoclast genes and proteins within the lungs and extracted microliths of PAM patients and PAM mouse model *Npt2b*^{-/-}, suggesting a potential role for osteoclasts in regulation of these CaP microliths. Implantation of microliths (adoptive transfer) into wild type mouse lungs induced osteoclast-like cells. Dietary phosphate regulated osteoclast-like cell formation in lungs, with low phosphate diet increasing osteoclast-like cells and ultimately reducing microliths in *Npt2b*^{-/-} mice. Inhibition of RANKL signaling via neutralizing antibody prevented the formation of TRAP positive osteoclast-like cells and prevented the clearance of microliths. Altogether, these results demonstrate a plasticity of lung myeloid/macrophage cells to exhibit an osteoclast-like phenotype in response to microlith accumulation. These findings have significant implications for treatment of PAM patients and provide novel therapeutic strategies. In addition, these results push forward our understanding of phosphate regulation of osteoclast phenotypes beyond the bone tissue. Overall, these are fascinating data and significant findings. Below I have noted specific instances in which the authors should provide additional information, alter methodology, or modify their interpretation.

Thank you for positive comments.

2a. It is not clear what lineage markers the authors used to define the macrophage population in the single cell analysis. This information should be provided to better understand how the population was selected.

Please see responses to **Critiques. 1d,f,g, 3a,b,c**. Extensive re-analyses of the single cell data were performed and are now shown in **Figs. 2 and 3** and **Supplementary Figs. 1,2,3,4**.

2b. Related to this question, were these markers specific to macrophages, or did they also include myeloid progenitor or dendritic cell lineages? Given that these lineages can all contribute to osteoclast differentiation in bone, it would be of interest to assess these populations as well (depending on whether they were already included in the original lineage selection).

To address this comment, we plotted the markers of distinct immune subtypes in a dot plot (**Supplementary Fig. 4**). Dendritic cells express RANK, ACP5 and multiple other macrophage markers, and may contribute to OC formation. Future lineage tracing experiments will address this question.

2c. Another related question is what happens to the macrophage marker F4/80 (ADGRE). Osteoclasts in bone are negative for this gene (while mononucleated osteomacs exhibit F4/80 and TRAP activity). Presence of F4/80 within this population would be valuable in assessing whether expression is reduced as these cells take on an osteoclast phenotype, similarly to the noted decrease in EMR1 expression in PAM myeloid cells.

To address this comment, we plotted ADGRE (official symbol: EMR1) on the integrated UMAP in the revision. Although F4/80 is expressed in both monocytes and macrophages, its expression is decreased in PAM macrophages compared to control macrophage cells (1.5 fold, p-value = $3.7E-2$). ACP5 (also known as TRAP) expression is significantly increased in PAM macrophages (fold change=1.8; p-value = $3.92E-32$). **Fig. 3b** shows the increased expression of ACP5 in PAM macrophages vs control. (It is our understanding that osteomacs are TRAP negative). Our conclusion is that there is a subtle but significant decrease of EMR1 in PAM macrophages, consistent with macrophage differentiation along an osteoclast pathway.

2d. Also, given the results in Figure 6, did the gene for RANK (TNFRSF11A) show up in this analysis?

Box plots and dot plots of expression of RANK (TNFRSF11A) and other osteoclast markers across ten immune cell types and across 5 PAM macrophage subtypes are shown in **Supplementary Figure 4 and Fig. 3d**. RANK expression is low but detectable in macrophage and dendritic cells, especially in PAM_Mac3 (**Fig. 3d**). In general, the osteoclast markers are more abundantly expressed in macrophages than in other immune subtypes (**Supplementary Figure 4**).

2e. Regarding the differential gene expression analysis of the single cell data, the methods state that the binomial nonparametric test was used, and differential expressed genes are $FDR < 0.1$, effect size > 2 , and frequency $> 20\%$. However in the figure legend this is only listed for the results in panel 2d. It is not clear whether the osteoclast genes are significantly increased from the figure and legend. Full differential gene expression results should be provided as supplemental data.

We used modified criteria for the group of osteoclast genes (p-value < 0.05 and effect size or fold change > 1.3) since some of these genes were only expressed in a subset of PAM macrophages. As suggested, the full DE genes were included as Supplemental table 5. We added clear descriptions of the criteria we used to select the osteoclast genes in the legend of the revision.

2f. In Figure 3g-r, it appears that the TRAP and CTSK positive cells in PAM lungs are multinucleated. However, the CALCR positive cells appear mononucleated. This raises the question of whether the CALCR positive cells are a separate population of myeloid cells. If possible, co-staining would be useful to determine if these signals colocalize, especially since the authors note CALCR expression as one of the defining osteoclast features in the discussion. **Please see new Fig. 4s, which demonstrates co-localization of TRAP and CALCR staining in both mononuclear cells and multinucleated giant cells.**

2g. The authors also mention in this section that multinucleated giant cells colocalized with microliths and were found adherent to microliths. From these images, it is not clear whether microliths are present. If so, it would be beneficial to indicate these with arrows. Interpretation

of immunostaining in Figure 3g-r would also be aided by labeling the figure directly to show which proteins are being identified in which panels.

Arrows indicating microliths were added to **panels 4h, i, j, n**. Staining with TRAP (**Fig. 4g-j**), CTSK (**Fig. 4k-n**) and CALCR (**Fig. 4o-r**) is indicated by labels on the left margin.

2h. The authors cultured BAL cells following intratracheal challenge with hydroxyapatite spheres on bovine bone chips to assess resorptive activity. It is unclear why the authors used hydroxyapatite in this assay instead of adoptive transfer of microlith or Npt2b^{-/-} mice, which are what they showed had increased osteoclast gene expression.

Microliths were used in experiments in **Figs. 4a-f** and **Figs. 5a-e**. Synthetic hydroxyapatite microspheres were used in **Figs. 5f-p**. Given that osteoclast differentiation is induced by both microliths and hydroxyapatite microspheres, and the expense and effort required to isolate microliths from Npt2b^{-/-} mice (one KO mouse yields enough stones for adoptive transfer to 1-2 mice), we felt that synthetic hydroxyapatite microspheres are a reasonable surrogate for natural microliths.

Given the additional components within the microliths, it would be preferable to at least include adoptive transfer of microliths or Npt2b^{-/-} mouse BAL cells as another group in this experiment. Alternatively, the authors should show that similarly to Npt2b^{-/-} mice (Figure 3a-e) or adoptive transfer of microliths (Figure 4a-d), the wild type hydroxyapatite recipient BAL cells express osteoclast markers.

In new experiments, we demonstrate that d14 BAL cells isolated from mice challenged with hydroxyapatite microspheres but not saline differentiate into TRAP⁺ osteoclast like MNGC when plated on plastic in the presence of low dose RANKL (25 ng/ml)(**Fig. 5f-p**). Similarly, d14 BAL cells isolated from mice challenged with hydroxyapatite microspheres but not saline form actin rings, pit bone and release CTX-1 into the media when plated on bovine femur slices in the presence of 100 ng/ml RANKL. For more detail, please see response 2l.

2i. The increase in CTX1 in the media is promising (Figure 3u), but does not confirm osteoclast resorption; this may simply represent the presence of increased protease activity. The authors should assess whether these BAL derived cells are forming actin rings that are crucial for formation of a resorption lacunae. Alternatively, the authors could stain the bone chips with toluidine blue to identify resorption pits following the culture period.

These experiments were completed and results are now shown in **Fig. 5l-o**.

2j. Previous studies have shown that CCR2^{-/-} bone marrow cultures have reduced osteoclast differentiation (PMID: 26205994). Thus, is it possible that the AM population has reduced plasticity towards osteoclast differentiation in CCR2^{-/-} mice? Do AM cells express CCR2?

AM cells do not express CCR2, as reported in: Opalek JM, Ali NA, Lobb JM, Hunter MG, Marsh CB. Alveolar macrophages lack CCR2 expression and do not migrate to CCL2. *J Inflamm (Lond)*. 2007 Sep 22;4:19. PMID: 17888174. This point is now included in the Discussion. We show in Supplementary Fig 6B that BAL cells from CCR2^{-/-} cultured in high dose (100 ng/ml) but now low dose RANKL + MCSF are able to form TRAP⁺ osteoclasts in culture.

2k. Osteoclast differentiation of bone marrow derived monocytes typically occurs in 4 days, with maximal osteoclast gene expression at day 3. After forming osteoclasts, these cultures

don't typically last longer than 5 days, as osteoclasts begin to undergo apoptosis. Given the sustained gene expression with microliths or hydroxyapatite, I'd be interested to see if these cells still look like osteoclasts and how much cell death has occurred. Overall, these results would also be more informative if TRAP staining images of the differentiation cultures at Days 3 and 7 were included.

In the interval since the first submission, we refined our osteoclast cell culture system. We obtained a proven lot of fetal calf serum from the Teitelbaum lab, and used media from CMG cells they provided us as an affordable, abundant source of MCSF to optimize viability. Cultures of MCSF conditioned BM cells incubated with 5-50 ng/ml RANKL demonstrated the formation of TRAP+ MNGC (see response 2l), with actin rings and bone pitting capacity, on day 6 of RANKL exposure and day 10 in culture (**Supplementary Fig. 6**). In experiments not shown, persistence of viable OCs was demonstrated out to day 11 of RANKL exposure, based on Live/Dead staining.

2l. Including images of multinucleated AM-derived osteoclasts would also be informative as to whether this cell population is able to differentiate into osteoclasts on their own, or whether they require recruitment of circulating monocytes. I would not necessarily conclude that CaP crystals potentiate osteoclast differentiation, unless there is a clear increase in TRAP positive multinucleated osteoclasts. If no quantification of TRAP staining can be performed, I would restate the sentence (lines 216-218) to specifically indicate osteoclast gene expression.

Under the refined osteoclast culture conditions outlined in 2k, we were not able to reproduce the effect of microliths or hydroxyapatite on OC viability in vitro, and these experiments were deleted from **Fig. 5**.

We instead conducted OC differentiation experiments using BAL derived AMs, which has not been previously reported to our knowledge (**Fig. 5 f-p**). BAL cells were isolated from animals that had been pretreated with synthetic hydroxyapatite (HA) microspheres or saline 2 weeks earlier. We demonstrated TRAP+ MNGC formation by BAL cells isolated from HA treated (**Fig. 5j**) but not saline pretreated (**Fig. 5g**) mice when plated on plastic in the presence of MCSF + low dose RANKL (25 ng/ml). At higher doses of RANKL (100 ng/ml), BAL cells HA treated mice formed many large MNGC(**Fig. 5k**), and those from saline pretreated mice also formed a few small TRAP+ MNGCs (**Fig. 5h**). We also demonstrated actin ring formation (**Fig. 5m**), bone pitting (**Fig. 5o**) and CTX-1 release into the media by BAL cells isolated from HA treated but not saline treated (**Fig. 5l,n, resp**) mice when plated on bovine bone slices in the presence of MCSF + 100 ng/ml RANKL. These data demonstrate that pretreatment of mice with i.t. HA enhances osteoclastogenesis in isolated d14 BAL cells.

2m. The concept of targeting osteoclasts to prevent fibrosis in PAM lungs is very interesting, and is somewhat analogous to the treatment of fibrous dysplasia patients with anti-RANKL therapy. It is possible that rather than being mediated by protease-induced damage, the fibrosis may also result from osteoclast-derived factors that contribute to tissue regeneration (termed coupling in bone remodeling).

An excellent point, thank you. The concept that that OCs could release TGF- β and IGF-1 from lung matrix, secrete clastokines, and directly interact with cells in the mesenchymal compartment to promote fibrosis is now included in the discussion.

Reviewer #3 (Remarks to the Author):

A review of the manuscript from Uehara et al., “Novel insights into pulmonary phosphate homeostasis and osteoclastogenesis emerge from the study of pulmonary alveolar microlithiasis”. PAM is a rare autosomal recessive lung disease in which microliths accumulate in the alveoli and is caused by mutations in the SLC34A2 gene which encodes a sodium-phosphate co-transporter. In this study, the authors expand on current data in terms of phosphate biology. In the pathogenesis of PAM they identify an osteoclast population/program derived from monocyte/macrophages in the PAM lung using a mouse (Npt2b^{-/-}) model and samples from an individual with PAM. They identify that the microliths induce osteoclast-like activity when transferred into healthy mice. Low phosphate diet increased the osteoclast-pattern of gene expression and was associated with increase clearance of microliths. A role for RANKL was identified in this process. High dietary phosphate led to elevated levels of osteoprotegerin (a decoy receptor from RANKL) and led to increased microlith burden. This study highlights the inability of the alveolar space to regulate phosphate metabolism in PAM and the identification of an osteoclast program opens new therapeutic research avenues. Multiple experiments are performed and summarized briefly here. First, a detailed analysis of PAM microliths (mice and one PAM patient) was performed and identified many proteins including osteoclast proteins. Osteopontin was identified as significantly elevated in PAM mice and patients (n=14). While statistically significant, the levels are variable in PAM patients. ScRNA-Seq of lung cells from a 2 year old PAM patient is performed vs a normal control sample (n=1 of each) and identified an osteoclast signature in PAM macrophages. Some of these genes were then validated in BAL cells. Multinucleated giant cells were identified, frequently adjacent to microliths, in PAM lungs and expressed TRAP, CTSK and CALCR. Enzymatically active osteoclast protease activity (CTSK) was demonstrated in BAL cells from Npt2b^{-/-} mice using an in vivo imaging system. Adoptive transfer of microliths into WT lungs was performed and led to increased expression of osteoclast genes. CCR2^{-/-} mice were used to determine the origin of cells with the osteoclast program and suggested monocytes are the major source and alveolar macrophages a minor source. These findings are further validated in an in vitro experiment culturing bone marrow derived monocytes of alveolar macs in osteoclast conditions with microliths suggesting the microliths potentiate the osteoclast program independent of mutations that cause PAM. Next the authors demonstrate that alterations in dietary phosphate concentration leads to changes in transport levels and phosphate/calcium levels in the alveolar space. High phosphate diet increase BAL phosphate and microliths however TRAP positive cells decreased. Phosphate transporter gene expression was decreased in AEC2 compared to low phosphate diet. In vivo imaging and RANKL neutralizing antibodies were used to demonstrate the role of RANKL in osteoclast formation using and assessing microlith clearance.

Although PAM is rare genetic disease the insights learnt by studying this disease would likely prove beneficial in understanding the pathobiology of other pulmonary diseases. Overall, this is a very well designed and rigorously conducted study that combines transcriptomic analyses, human samples, mouse genetic models, and pharmacologic therapies. In my opinion, the work

by Uehara and colleagues meets Nature Communications high publication standards and therefore recommend to accept with minor revisions.

Thank you for positive comments.

Major:

3a. Fig. 2: the scRNA-Seq of a single human sample is key to the papers findings. Further data analysis of this scRNA-Seq would add to the paper. The authors focus on the differences in gene expression in lung macrophage populations between a patient and a control explant. Since scRNA-seq analysis of whole lungs was performed, it would be of interest to the reader to first show a combined UMAP of all cells in the patient and control explants and to annotate all cell clusters.

Extensive reanalysis of single cell data was performed to address concerns of all reviewers. See Fig. 2 and 3, and Supplementary figs 1,2,3,4.

3b. It is also unclear to this reader whether the osteoclast-related genes were among the top differentially upregulated genes. Including a heatmap of the top differentially genes in patient macrophages vs control would be useful.

This was done. Please see new Fig. 3a.

Furthermore:

3c. although only n=1, are there differences in the AEC2 in the PAM patient (inflammatory, pro-fibrotic pathways?)

To address this question, we compared differentially expressed genes in AT2 cells in PAM vs control. We observed that genes involved in the bioprocesses of "protein translation", "intracellular protein transport," and "cytokine secretion" were upregulated. Down-regulated genes were enriched in "lipid/cholesterol biosynthesis/metabolism", "unfolded protein response (UPR)", "ATP synthesis coupled electron transport" and "surfactant metabolism/transport". ABCA3, SFTPA2, SFTPA1, SFTPD, SLC34A2, COX6A1 and COX7B were downregulated in PAM AT2 cells. Pro-fibrotic pathways such as "Jak-STAT" and "TNF signaling" were induced in PAM AT2 cells.

As the reviewer suspected, the PAM epithelial cells show very interesting findings but we felt inclusion might be distracting since the entire manuscript focuses on myeloid lineages. We plan to complete a more systematic assessment and functional validation in the

future using single cell RNA seq mouse models (since the likelihood of obtaining another human PAM lung sample is small). Here we attached functional enrichment results of AT2 cell comparisons, but were not planning include this in the current manuscript.

3d. what is the expression of the transporters described later in the human samples

SLC34A1, SLC34A3, S100g were not detected in the PAM lung. SLC20A1 and SLC20A2 were decreased in the PAM lung relative to control. XPR1 and TRPV6 were marginally increased in the PAM lung relative to control, but the level of expression was below the cutoff line (10%) and the results are not reliable. The most relevant question is whether these genes are regulated by dietary phosphate intake, and we relied on rtPCR to address this question (**Fig. 6s,t and Supplementary Figure 8**).

3e. were MNGCs included in the scRNA seq or would these be excluded by the sample preparation of bioinformatic analysis

There appear to be two populations deemed macrophages in the scRNA seq of human lungs, perhaps more in the PAM patient. This heterogeneity is not addressed. Can the authors clarify the nature of these clusters and provide data if needed demonstrating the differences. The expression of canonical macrophage/myeloid markers in a supplemental figure would be useful in understanding this potential heterogeneity and its relevance to the osteoclast gene expression program.

We agree that MNGC were likely excluded by sample preparation (now included Discussion section as a limitation). We do find expression of OC markers (CALCR and TRAP) in BAL cells and in mononuclear cells by IHC. The single cell analysis was extended to include delineation of macrophage subtypes and these data are now included in **Fig. 3c-e and Supplementary Fig 3**.

3f. Are there “normal’ macrophage in the PAM scRNA-seq data?

Yes, please see response **1f**. The integrated UMAP of AM from PAM vs control in **Figs. 3 and Supplementary Fig. 2**. shows significant overlap, consistent with presence of ‘normal AM’ in the PAM lung

3g. Are there are subset of cells that are most osteoclast-like Louvain clustering (or some alternative) applied to the myeloid clusters and violin plots of the osteoclast genes shown in 2b and c would provide more granular view of their expression.

In the revision, macrophage cells were reanalyzed using the Jaccard-Louvain clustering algorithm (**Fig. 3c-e and Supplementary Fig. 1 and 3**). PAM-Mac3 is the most osteoclast-like cell cluster . Expression of osteoclast markers among different myeloid cells were visualized in box plots in the revision (**Supplementary Fig. 4**).

3h. Within the PAM myeloid cells, is there any evidence to suggest differentiation trajectories from AMs/monocytes to osteoclasts.

To address reviewer's comments, we conducted the Monocle3 trajectory analysis using combined PAM monocytes and PAM macrophage subtypes. As shown in **Fig. 3e** the predicted trajectory started from monocytes →PAM-Mac4 (monocyte-macrophage fusion)→PAM-Mac3 (osteoclast-like). PAM-Mac4 also serves as a progenitor of other macrophage branches (PAM-Mac1, 2 and 5).

3i. . This is a rare disease thus understandable that scRNA-Seq was only feasible on one patient. However, if key findings could be validated, e.g. paraffin fixed sections, in other patients this would increase confidence in the human findings.

We obtained 4 archival PAM lung paraffin sections from colleagues in Aarhus, Denmark; Rochester MN; Osaka, Japan; and Houston, TX. Although all were quite old and in poor condition, we were able to identify TRAP (red) and CTSK (brown) positive mononuclear and MNGC cells surrounding stones in the Denmark sample and TRAP (red) and CALCR (brown) positive mononuclear and MNGC cells surrounding stones in the Mayo samples. The Osaka sample was corrupted/unusable and there was little tissue with few microliths in the Houston sample. We have made note of the additional patient samples in text and now show the IHC from them in Supplementary Fig. 5.

Minor:

3j. The link to osteopontin biomarker studies is clear from the figures but not the text.

-The conclusion that osteopontin is expressed in AEC1 and AEC2 cells is not well supported by the immunostaining which would be improved by co-staining with canonical markers and higher resolution images (Fig 1o). Additional patient samples, if available, would strengthen this conclusion.

Based on reanalysis of the single cell data, >95% of SPP1 expression in the PAM lung was found to be in myeloid lineage cells (See Fig 2a for cell annotations), and we focused on that subset. Please see response 4a below. We did not further pursue IHC analysis of epithelial OPN expression.

3k. Details of PAM patient mutation

This patient had a homozygous mutation in c.524-18_559del. This is now included in the text.

3m. Line 143-144: The term infant is used for ages 0-1 years; consider using an alternative term. We now refer to the patient as the PAM child.

3n. Line 144: "A heatmap illustrates upregulation of expression of key osteoclast related genes in PAM lung macrophages (Fig.2b-e)" should be revised to more accurately describe the different plots used in 2b-e

Please see revised **Figs. 2 and 3.**

3o. Fig 1n: significant variation in serum osteopontin. Do the two outliers have worse disease, and are they the outliers in Fig5 g-i

Unfortunately, these samples were collected from all over the world (Turkey, Japan, US) in a deidentified manner. We don't have the linked clinical data that would be required to assess disease severity.

-Line 143-144: The term infant is used for ages 0-1 years; consider using an alternative term.

This point was duplicated by the reviewer. See 3m.

Reviewer #4 (Remarks to the Author):

This manuscript is an overall well-written investigation of a rare lung disease, pulmonary alveolar microlithiasis (PAM), and the role of Npt2b in pulmonary phosphate homeostasis and microlith clearance. This work leverages scRNA-seq technology in an explanted diseased lung as well as murine models, including dietary interventions, to argue for a critical role for Npt2b and osteoclast-like pulmonary cells in phosphate homeostasis, which may support future discovery of therapeutics for this disease. While interpretations of the analysis are certainly limited by sample number, this would be the first published scRNA-seq analysis of PAM lung, and will provide important data for the scientific community as this is a rare disease with limited availability of patient tissue. The work could be improved by further details and attention to a few particular areas:

Thank you for positive comments.

4a. Osteopontin is suggested as a potential biomarker based on its elevated serum levels and murine BALF levels. Additionally, IHC staining suggested it localized to AECI, AECII, and microlith adjacent myeloid cells in the PAM lung. How does this expression correlate with SPP1 (osteopontin) expression in the scRNA-seq data? I.e which cell types are predominantly expressing this and does that change from control to disease? For instance, scRNA-seq data in multiple ILDs has demonstrated a distinct population of macrophages with high expression of SPP1 that are expanded in fibrosis compared to controls, and suggest that macrophages, not alveolar epithelial cells, are the predominant source of osteopontin in fibrosis.

Very helpful, thank you. Indeed, SPP1 was predominantly expressed in macrophages and the percent of AM that were SPP1 positive was more than 3-fold higher in PAM vs control. SPP1 was expressed at a low level in control AT2 and ciliated bronchial epithelial cells but was not detected in control AT1 cells. Comparing PAM AT2 to control, OPN expression in AT2 did not change, increased marginally in ciliated bronchial epithelial cells and became detectable in AT1 cells, but collectively epithelial cells represented less than 5% of all OPN positive cells in the PAM lung (now noted in the text).

4b. Additional data should be provided on the scRNA-seq analysis methods including how the lung samples were digested, 10X Genomics chemistry used, specific method for combining the samples, batch correction.

As suggested, we now provided these details in the method section of the revision.

4c. Please include UMAPs of the lung sample overall clusterings identifying all cell types, as well as demonstration of gene expression for how the macrophages were identified.

Please see revised **Figs 2 and 3**, and **Supplementary Figs. 1,2,3,4**

4d. Differential expression testing results for the "upregulated key osteoclast related genes", or all genes between the PAM and control macrophages needs to be included rather than visualizations alone. Currently how the depicted genes were chosen and the level of significance for them is unclear.

As suggested, the full DE genes were included as **Supplementary table 5** in the revision. We added clear descriptions of the criteria we used to select the osteoclast genes in the legend of the revision. For details, please see response to 2e.

4e.. In Figure 2E, a DotPlot may be a more informative by showing both the percentage of cells expressing each gene and the expression level across both the disease and control, rather than the percent expressing alone.

In the revision, we replaced the heatmap by a dot plot (**Fig. 3b**) to show both the percentage of cells expressing each gene and the expression level across both the disease and control.

4f. Are specific subgroups of macrophages able to be identified to suggest a more precise origin for the osteoclast like cells? Downregulation of a single gene, EMR1, is insufficient to suggest that there is osteoclast differentiation in AM of the PAM lung.

This was a common concern of multiple reviewers. The heterogeneity of myeloid populations is addressed by sub-clustering of macrophages in PAM and control lung (**Fig. 3**). In addition to EMR1 (Fig 2d), we plotted a panel of osteoclast markers in different macrophage subtypes and different myeloid cells in the revision (**Fig. 3c-e, Supplementary Figs.3, 4**). PAM-Mac3 is the most osteoclast-like cell, likely derived from PAM-Mac4 (monocyte-macrophage fusion) (**Fig. 3d, e**).

4g. In Figure 5, the authors show compelling effects of dietary phosphate on calcific densities and stone burden in the lung. Do the authors have any further phenotyping of these mice such as physiologic lung function to know if this difference in stone burden translated to a meaningful physiologic effect? Were there any significant changes in weight, mortality, and/or other indicators of murine health with the dietary intervention?

We did not measure differences in weight or physiologic function in mice treated with various phosphate diets. We did not observe any mortality in these experiments. The mice on low and high phosphate diets did not appear to be ill.

4h. In Figure 5 panels d-e change in stone burden is shown only for the 0.7% and 0.1% Pi diet. This information should be included for the 2.0% Pi diet as well.

These data are now included in **Fig. 6d, e, f and Supplementary Fig.7**.

4i. In Figure 3 panels g-r, visual interpretation would be easier by including the stains (TRAP, CTSK, CALCR) on the left y axes, similar to the tissue type labeling above the figure.

This was done.

4j. There is no discussion of limitations of the work. This should be thoughtfully considered and included within the discussion section.

This section was added to the discussion.

4k. Manuscript does not include information on data availability, where sequencing data will be deposited, etc.

This information is now included in the methods section.

REVIEWERS' COMMENTS

Reviewer #1 (Remarks to the Author):

dear authors,

well done. you have addressed most of my concerns and improved the study. i accept that some of my suggestions might have been beyond the scope of this paper.

taken together i am happy to see this study being published in ncomms.

Reviewer #3 (Remarks to the Author):

I find the revised manuscript responsive to the input of the reviewers. My concerns/questions have been addressed. In particular, I find the additional scRNA-seq analysis improves the manuscript.

Again, while $n=1$ for the scRNA-Seq is not ideal the reviewers have used additional human samples to validate key findings in addition to the mouse model. I find the data presented here to be novel and of significant interest to the lung community.

There are some minor errors in the text:

-Line 175 "a)" missing?

-Define MNGC at first use

F. Hawkins

Reviewer #4 (Remarks to the Author):

The authors have adequately addressed all of my concerns and greatly improved the quality and findings of the manuscript with this revision.

REVIEWERS' COMMENTS

Reviewer #1 (Remarks to the Author):

dear authors,

well done. you have addressed most of my concerns and improved the study. i accept that some of my suggestions might have been beyond the scope of this paper.

taken together i am happy to see this study being published in ncomms.

We appreciate the helpful comments.

Reviewer #3 (Remarks to the Author):

I find the revised manuscript responsive to the input of the reviewers. My concerns/questions have been addressed. In particular, I find the additional scRNA-seq analysis improves the manuscript. Again, while n=1 for the scRNA-Seq is not ideal the reviewers have used additional human samples to validate key findings in addition to the mouse model. I find the data presented here to be novel and of significant interest to the lung community.

We appreciate the helpful comments.

There are some minor errors in the text:

-Line 175 "a)" missing?

We added "a)" in Line 175.

-Define MNGC at first use

We defined first MNGC.

F. Hawkins

Reviewer #4 (Remarks to the Author):

The authors have adequately addressed all of my concerns and greatly improved the quality and findings of the manuscript with this revision.

We appreciate the helpful comments.